 TOOLS AND RESOURCES

# Data-driven reduction of dendritic morphologies with preserved dendro-somatic responses

Willem AM Wybo*, Jakob Jordan, Benjamin Ellenberger, Ulisses Marti Mengual, Thomas Nevian, Walter Senn

Department of Physiology, University of Bern, Bern, Switzerland

**Abstract** Dendrites shape information flow in neurons. Yet, there is little consensus on the level of spatial complexity at which they operate. Through carefully chosen parameter fits, solvable in the least-squares sense, we obtain accurate reduced compartmental models at any level of complexity. We show that (back-propagating) action potentials, $Ca^{2+}$ spikes, and $N$-methyl-D-aspartate spikes can all be reproduced with few compartments. We also investigate whether afferent spatial connectivity motifs admit simplification by ablating targeted branches and grouping affected synapses onto the next proximal dendrite. We find that voltage in the remaining branches is reproduced if temporal conductance fluctuations stay below a limit that depends on the average difference in input resistance between the ablated branches and the next proximal dendrite. Furthermore, our methodology fits reduced models directly from experimental data, without requiring morphological reconstructions. We provide software that automatizes the simplification, eliminating a common hurdle toward including dendritic computations in network models.

## Introduction

Morphological neuron models have been instrumental in neuroscience (*Segev and London, 2000*). Major experimental discoveries, for instance that $N$-methyl-D-aspartate (NMDA) channels (*MacDonald and Wojtowicz, 1982*) produce local dendritic all or none responses (*Schiller et al., 2000*; *Major et al., 2008*), or that dendritic $Ca^{2+}$ spikes mediate coincidence detection between distal inputs and somatic action potentials (APs) (*Larkum et al., 1999*), have been combined in morphological models to arrive at a consistent picture of dendritic integration: the dendrite is an intricate system of semi-independent subunits (*Mel, 1993*; *Poirazi et al., 2003b*; *Poirazi et al., 2003a*), amenable to dynamic regulation (*Poleg-Polsky et al., 2018*; *Wybo et al., 2019*), and able to distinguish specific input patterns (*Branco and Häusser, 2010*; *Laudanski et al., 2014*).

Nevertheless, morphological models are not without shortcomings. They are highly complex and consist of thousands of coupled compartments, each receiving multiple non-linear currents. The parameters of the models, typically fitted with evolutionary algorithms to electro-physiological recordings (*Hay et al., 2011*; *Almog and Korngreen, 2014*; *Van Geit et al., 2016*), number in the tens of thousands. Since recordings can only be obtained at a few dendritic sites, these fits are under-constrained, and thus susceptible to over-fitting. Accordingly, the single-neuron fitting challenge, where model performance was measured on unseen spike trains, was not won by a biophysical model, but by an abstract spiking model (*Gerstner and Naud, 2009*; *DiLorenzo and Victor, 2013*). Finally, many network-level observations can be explained without morphological models (*Gerstner et al., 2012*).

These shortcomings underline the need to find the essential computational repertoire of a neuron: the set of computations needed to understand brain functions such as learning and memory. Model simplification is crucial in this endeavor, as it elucidates the lowest level of complexity at

*For correspondence:
willem.a.m.wybo@gmail.com

Competing interests: The authors declare that no competing interests exist.

which computational features are preserved. Conceptually, the simplification thus extracts the essential elements required for the computation from the underlying biophysics. Simulation-wise, the reduced model requires few resources and can thus be integrated in large-scale networks. Experimentally, the reduced model likely leads to a well-constrained fit.

Past simplification efforts can be grouped in two categories: approaches that use traditional compartments, but with adapted parameters (*Pinsky and Rinzel, 1994*; *Destexhe, 2001*; *Tobin et al., 2006*; *Bahl et al., 2012*; *Marasco et al., 2013*; *Amsalem et al., 2020*), and approaches that rely on more advanced mathematical techniques (*Kellems et al., 2009*; *Kellems et al., 2010*; *Wybo et al., 2013*; *Wybo et al., 2015*). Many authors apply ad hoc morphological simplifications and then adjust model parameters to conserve geometrical quantities, such as surface area (*Davison et al., 2000*; *Hendrickson et al., 2011*; *Marasco et al., 2013*), or electrical quantities, such as attenuation (*Destexhe, 2001*) and transfer impedance (*Amsalem et al., 2020*). Other authors propose two-compartment models whose parameters are adjusted to reproduce certain response properties (*Pinsky and Rinzel, 1994*; *Naud et al., 2014*). Nevertheless, these approaches often lack the flexibility to be adapted to a wide range of dendritic computations. Furthermore, particular afferent spatial connectivity motifs might be lost, so that essential computations are not captured by such a reduction. Hence, in a network model, the computational relevance of these motifs might unknowingly be ignored. Advanced mathematical techniques on the other hand are very flexible, as they can incorporate the response properties of the morphology implicitly. (*Wybo et al., 2013* and *Wybo et al., 2015* propose a system of convolutions whose kernels capture the passive properties of the compartmental model, whereas *Kellems et al., 2009* and *Kellems et al., 2010* linearly transform the compartmental model into a low-dimensional basis.) However, these techniques are not supported by standard simulation software, such as NEURON (*Carnevale and Hines, 2004*), a considerable hurdle toward their integration in the canonical neuroscience toolset.

Here, we introduce a simplification method with the flexibility of advanced mathematical techniques, but using traditional compartments. The approach accommodates any morphological model in public repositories (*McDougal et al., 2017*) and commonly used ion channels (*Podlaski et al., 2017*). The reduced models represent the optimal approximation to the dendritic resistance matrix, evaluated at any spatial resolution of choice. We fit ion channels by approximating the quasi-active resistance matrices (*Koch and Poggio, 1985*). All our fits are uniquely solvable linear least-squares problems. The obtained models extrapolate well to non-linear dynamics, and reproduce back-propagating APs (bAPs), $Ca^{2+}$ spikes (*Larkum et al., 1999*), and NMDA spikes (*Schiller et al., 2000*; *Major et al., 2008*) with few compartments. Additionally, we investigate whether a dendritic tree with given afferent spatial connectivity motifs can be simplified by ablating subtrees or branches and grouping synapses in the next proximal compartment. We find that effective weight-rescale factors for synapses can be computed if temporal conductance fluctuations stay below a limit that depends on the difference in input resistance between the ablated branch and the next proximal dendrite. Under application of these factors, voltage responses in the simplified tree are preserved. Finally, we demonstrate that our approach can fit reduced models to experimental recordings, without the need to reconstruct full morphologies. We have created a Python toolbox (NEural Analysis Toolbox – NEAT – https://github.com/unibe-cns/NEAT) (copy archived at swh:1:rev:1cb15f36aa0a764105348541d046c85ef38e21ee) that implements this method together with an extension to NEURON to simulate the obtained models.

## Results

### A systematic simplification of complex neuron morphologies

Our simplification strategy fits compartments with voltage-gated channels to a reduced set of dendritic locations of interest (*Figure 1A*, left-middle). The locations, together with the morphology, provide a tree structure for the reduced model (*Figure 1A*, middle-right) where each node corresponds to a traditional compartment and each edge to a coupling conductance. The fit requires that the reduced model responds similarly to perturbative current steps as the full neuron, at the set of chosen locations (*Figure 1B*). We consider perturbations around any spatially uniform holding potential $v_h$. Experimentally, $v_h$ may be reached by injecting a constant current $i_h$, and thus $v_h$ is the effective equilibrium potential under this current injection. In models, we do not need to know $i_h$; we

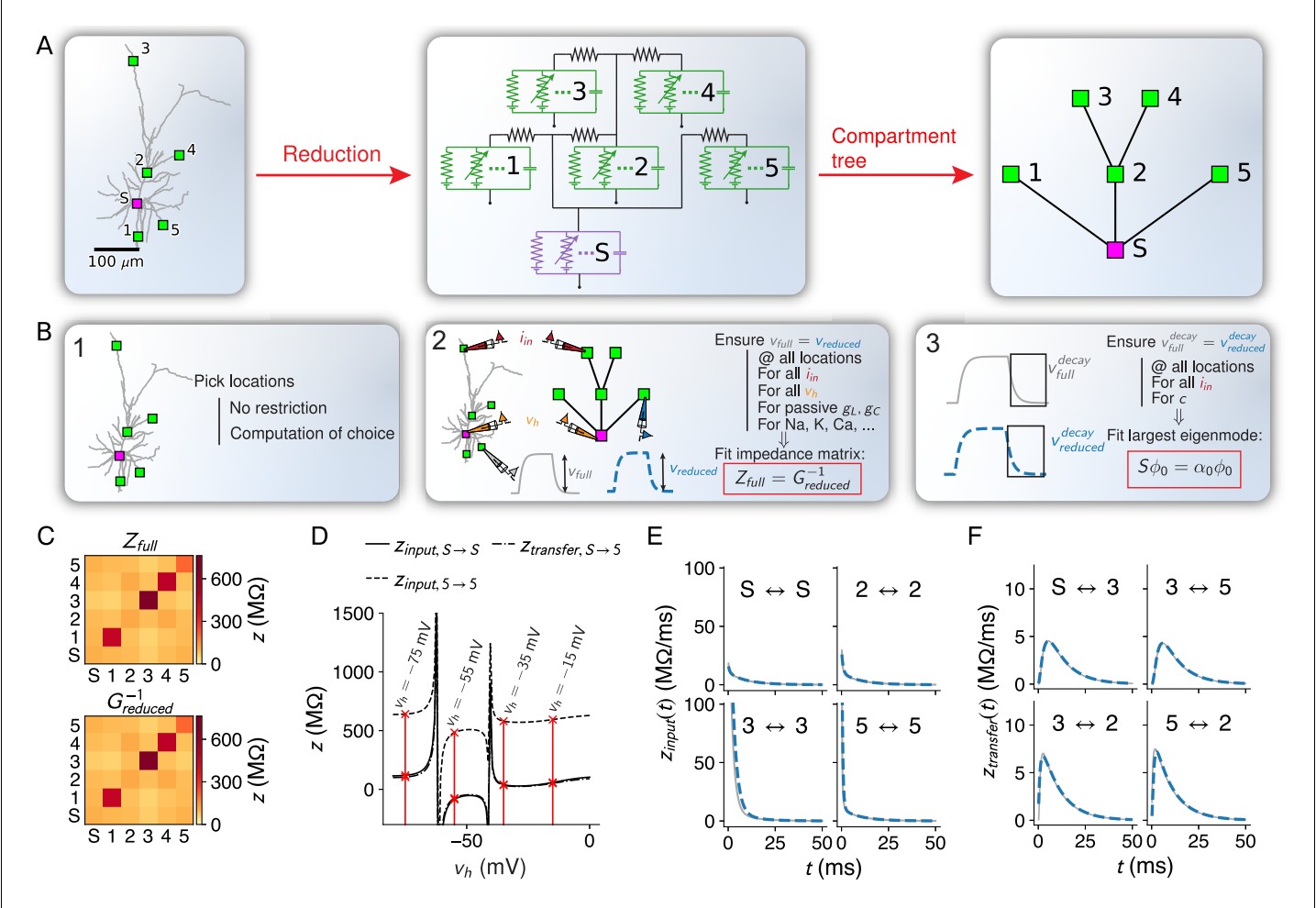

**Figure 1.** Flexible and accurate reduction methodology. (A) For any set of locations on a given morphology (left, here an L2/3 pyramidal cell [*Branco and Häusser, 2010*]), a reduced compartmental model can be derived (middle), with an associated schematic representation (right). (B) Steps of our approach: (1) choice of locations at which the reduced model should reproduce the full model's voltage, (2) coupling, leak and channel conductances are fitted to resistance matrices derived from the full model at different holding potentials, and (3) capacitances are fitted to mimic the largest eigenmode of the full model. (C) The resistance matrix of the passive full model (top) restricted to the five locations in A is approximated accurately by the inverse of the conductance matrix of the passive reduced model (bottom). Labels correspond to locations in A. (D) Example components of the quasi-active resistance matrix of the full model, equipped with a Na$^+$-channel, as a function of the holding potential $v_h$. Red lines show the four holding potentials at which our methodology evaluates the resistance matrix. Singularities correspond to holding potentials where the linearization is invalid and should be avoided in the fit. Labels correspond to locations in A. (E) Temporal shape of exemplar input impedance kernels of the full model (gray) and their reduced counterparts (blue, dashed). (F) Same as in E, but for transfer impedance kernels.

The online version of this article includes the following figure supplement(s) for figure 1:

**Figure supplement 1.** Resistance matrix fit details.

simply assume that $v_h$ is the equilibrium potential around which to linearize the model. Note that if $i_h = 0$, $v_h$ is the resting membrane potential.

Suppose we have chosen $N$ locations. Let $\delta\mathbf{i}$ be a vector describing perturbative input currents to each of those $N$ locations. The linearized voltage response of the full neuron, at those locations, is given in the Fourier domain for an input of a particular frequency $\omega$ by

$$\delta\mathbf{v}(\omega) = Z_{v_h}(\omega)\,\delta\mathbf{i}(\omega), \tag{1}$$

with $Z_{v_h}$ the quasi-active (*Koch and Poggio, 1985*) $N \times N$ impedance matrix. In experiments, $Z_{v_h}$ is extracted from $\delta\mathbf{v}$ measured in response to $\delta\mathbf{i}$. In models, $Z_{v_h}$ is algorithmically computed based on the full morphology, the parameters of the passive currents, and the dynamics of the voltage-gated

ion channels (*Koch and Poggio, 1985*). For our reduction method, we consider perturbative current steps, so that after an equilibration period, we obtain a steady-state voltage response $\delta\mathbf{v}$. Note that this corresponds to evaluating (*Equation 1*) at zero frequency, and that the corresponding matrix $Z_{v_h}(\omega = 0)$ contains the input and transfer resistances. Throughout the manuscript, we maintain the notation that $z$ without argument signifies $z(\omega = 0)$, representing an individual input or transfer resistance, and that $Z$ signifies $Z(\omega = 0)$, representing the resistance matrix.

Linearizing the reduced compartmental model around a holding potential (see Materials and methods — The conductance matrix) yields an $N \times N$ conductance matrix $G_{v_h}$, with on its diagonal

$$g_{Li} + \sum_{n \in \mathcal{N}_i} g_{C,in} + \sum_{d \in \mathcal{I}_i} \overline{g}_{di} l_d(v_h), \tag{2}$$

where $g_{Li}$ is the unknown leak conductance of the $i$-th compartment, $\mathcal{N}_i$ the set of nearest neighbor compartments, and $g_{C,in}$ the unknown coupling conductance between compartment $i$ and compartment $n \in \mathcal{N}_i$. The second sum runs over the set $\mathcal{I}_i$ of all linearized currents of ion channels present in compartment $i$, where $\overline{g}_{di}$ is the unknown maximal conductance of ion channel $d$ and $l_d(v_h)$ is a factor that follows from linearizing the dynamics of channel $d$ around the holding potential $v_h$ (see Materials and methods – Quasi-active channels). Thus, the unknowns $g_{Li}$, $g_{C,in}$ for $n \in \mathcal{N}_i$, and $\overline{g}_{di}$ for $d \in \mathcal{I}_i$ are to be fitted for each compartment, and the factors $l_d(v_h)$ are known and determined by the ion-channel dynamics. The $ij$'th off-diagonal element of $G_{v_h}$ is $-g_{C,ij}$ – the negative of the coupling conductance between compartments $i$ and $j$ – if $i$ and $j$ are nearest neighbor compartments on the reduced tree structure, and zero otherwise. The conductance matrix relates the reduced model's voltage response to the perturbative input current steps

$$G_{v_h} \delta\mathbf{v} = \delta\mathbf{i}. \tag{3}$$

The full neuron and reduced model will behave similarly for all possible perturbative input steps $\delta\mathbf{i}$ if their responses $\delta\mathbf{v}$ match. From (*Equation 1*) and (*Equation 3*), it follows that $Z_{v_h}$ should be the inverse of $G_{v_h}$. Consequently, we require that multiplying the known $Z_{v_h}$ (measured or calculated) by the parametric $G_{v_h}$ yields the identity

$$Z_{v_h} G_{v_h} \approx I. \tag{4}$$

From (*Equation 2*), it can be seen that (*Equation 4*) is linear in the parameters that have to be fitted (leak, coupling, and maximal ion-channel conductances of the reduced compartments). By consequence, (*Equation 4*) can be cast into a least-squares problem and solved accurately (*Figure 1C*, *Figure 1—figure supplement 1A*).

Since the linearized ion channel activation $l_d(v_h)$ depends on the holding potential, $Z_{v_h}$ changes with $v_h$ (*Figure 1D*). The fit must disentangle the changes in $Z_{v_h}$ induced by the various channels. In models, we first block all voltage-gated ion channels in the full and reduced models and fit $G_{\text{pas}}$ to $Z_{\text{pas}}$ according to (*Equation 4*), and thus obtain leak and coupling conductances for each compartment. Then, we unblock one ion channel at a time and decompose the conductance matrix into $G_{v_h} = G_{\text{pas}} + G_{v_h,\text{chan}}$, with $G_{v_h,\text{chan}}$ a diagonal matrix containing the conductances of the unblocked channel, linearized around $v_h$, at each compartment. Thus $G_{v_h,\text{chan}}$ depends linearly on the unknown maximal ion channel conductance parameters. With $Z_{v_h}$ and $G_{\text{pas}}$ known, we optimize these maximal conductances, so that the left-hand side of

$$Z_{v_h} G_{v_h,\text{chan}} \approx I - Z_{v_h} G_{\text{pas}}. \tag{5}$$

matches the right-hand side, and that it does so for multiple holding potentials (we chose $v_h = -75, -55, -35$, and $-15$ mV, *Figure 1D*, *Figure 1—figure supplement 1D*). Thus, we obtain an overdetermined system of equations in the unknown maximal channel conductances at each compartment and compute its solution in the least mean squares sense.

Having fixed the steady-state behavior of the reduced model, we are left with the capacitance of each compartment of the reduced model to match the temporal dynamics of the full model as closely as possible. In our reductions, the capacitance of each compartment is thus a parameter to be fitted. To do so, we found it sufficient to consider passive membrane dynamics. Blocking all voltage-gated ion channels in the reduced model ($G_{v_h,\text{chan}} = 0$), temporal voltage responses follow

$$\text{diag}(\mathbf{c})\dot{\mathbf{v}} = -G_{\text{pas}}\mathbf{v} + \mathbf{i}, \tag{6}$$

with $\mathbf{c}$ a vector containing the unknown capacitance of each compartment. We require that voltage-decay back to rest, as described by (*Equation 6*), matches voltage decay in the full neuron with all active channels blocked, at the $N$ chosen locations. According to linear dynamical systems theory (*Strogatz, 2000*), this decay can be decomposed as a sum of exponentially decaying eigenmodes, each with an associated eigenvalue $\alpha_k$ and eigenvector $\phi_k$. The former gives the exponential time scale $\tau_k = 1/\alpha_k$ of the decay and the latter the spatial profile of the mode. The most prominent of these modes has the largest time scale $\tau_0 = 1/\alpha_0$, and primarily models the voltage decay back to rest through trans-membrane currents (*Holmes et al., 1992*). In experiments, we extract this mode by fitting an exponential to the voltage decay. In full models, we compute the eigenvalue $\alpha_0$ and eigenvector $\phi_0$ – restricted to the $N$ chosen locations – with the separation-of-variables method (*Major et al., 1993*). In the reduced model, eigenmodes are found as the eigenvalues and eigenvectors of the matrix $S = -\text{diag}(\mathbf{c})^{-1}G_{\text{pas}}$. To fit the capacitances, we require that $S$ has $\alpha_0$ as eigenvalue and $\phi_0$ as corresponding eigenvector:

$$S\phi_0 = \alpha_0\phi_0. \tag{7}$$

This system of equations is linear in the reciprocals of the capacitances entering in $S$ and can thus be solved efficiently.

To accurately reproduce the spatio-temporal voltage responses of the full model, the reduced model must approximate the 'impedance kernels' $z(t)$ (*Wybo et al., 2015*), the inverse Fourier transforms of the frequency-dependent elements of the impedance matrix. Despite only fitting the largest time scale $1/\alpha_0$, we accurately reproduce kernels at all times (*Figure 1E,F*, *Figure 1—figure supplement 1B*).

Unless a large number of closely spaced compartment sites is retained on the morphology, our reductions replace the true axial currents flowing along the dendritic tree, which are shaped by the distribution of ion channels in the dendritic membrane, with a strong abstraction: a single coupling conductance between compartments that may be far apart. Hence, it is not guaranteed that the resting voltage of reduced model will be equal to the resting voltage of the full model, at the $N$ chosen locations. We ensure that this will be the case by fitting the leak reversals for each compartment. The product of leak conductance and reversal represents a constant current for each compartment, and thus does not influence model dynamics beyond fixing it's resting voltage. To fit the leak reversals, we evaluate all ion-channel currents in the reduced model at the full model's resting voltage and require that temporal voltage derivatives are zero. The obtained equations are linear in the reduced model's leak reversals, and hence can be solved efficiently. Note that our methodology does not guarantee that the obtained reversals are within physiological limits. Thus, they should purely be seen as fit parameters that determine the resting membrane potential.

## Reduced models match the voltage response of their full counterparts

We demonstrated the reduction on two computations that require accurate spatio-temporal interaction. We reproduced sequence discrimination in an L2/3 pyramidal cell model (*Branco and Häusser, 2010*), where a neuron responds more strongly to a centripetal sequence of inputs than to a centrifugal one, by only retaining compartments on a single branch (*Figure 2A*). We also reproduced input-order detection (*Torben-Nielsen and Stiefel, 2009*), where a neuron responds more strongly when one input arrives before the other, by only retaining compartments at the soma and the two input sites (*Figure 2B*).

We then tested our approach with non-linear currents concentrated at discrete points (hot-spots) along the morphology. In two biophysical models – an L5 neocortical pyramid (*Hay et al., 2011*; *Figure 2C*) and a Purkinje cell (*Chen et al., 2013*; *Miyasho et al., 2001*; *Figure 2D*) – we removed active dendritic channels, but added their total conductance at rest to the passive leak (we term this the 'passified' model), while retaining all active channels at the soma. The soma, with its AP-generating channels, naturally forms such a non-linear hot-spot. We implemented dendritic hot-spots by clustering sets of excitatory (α-amino-3-hydroxy-5-methyl-4-isoxazolepropionic acid [AMPA] + NMDA) and inhibitory (γ-aminobutyric acid [GABA]) synapses at randomly selected points on the morphology. Both the reduced pyramidal cell model and the reduced Purkinje cell model accurately

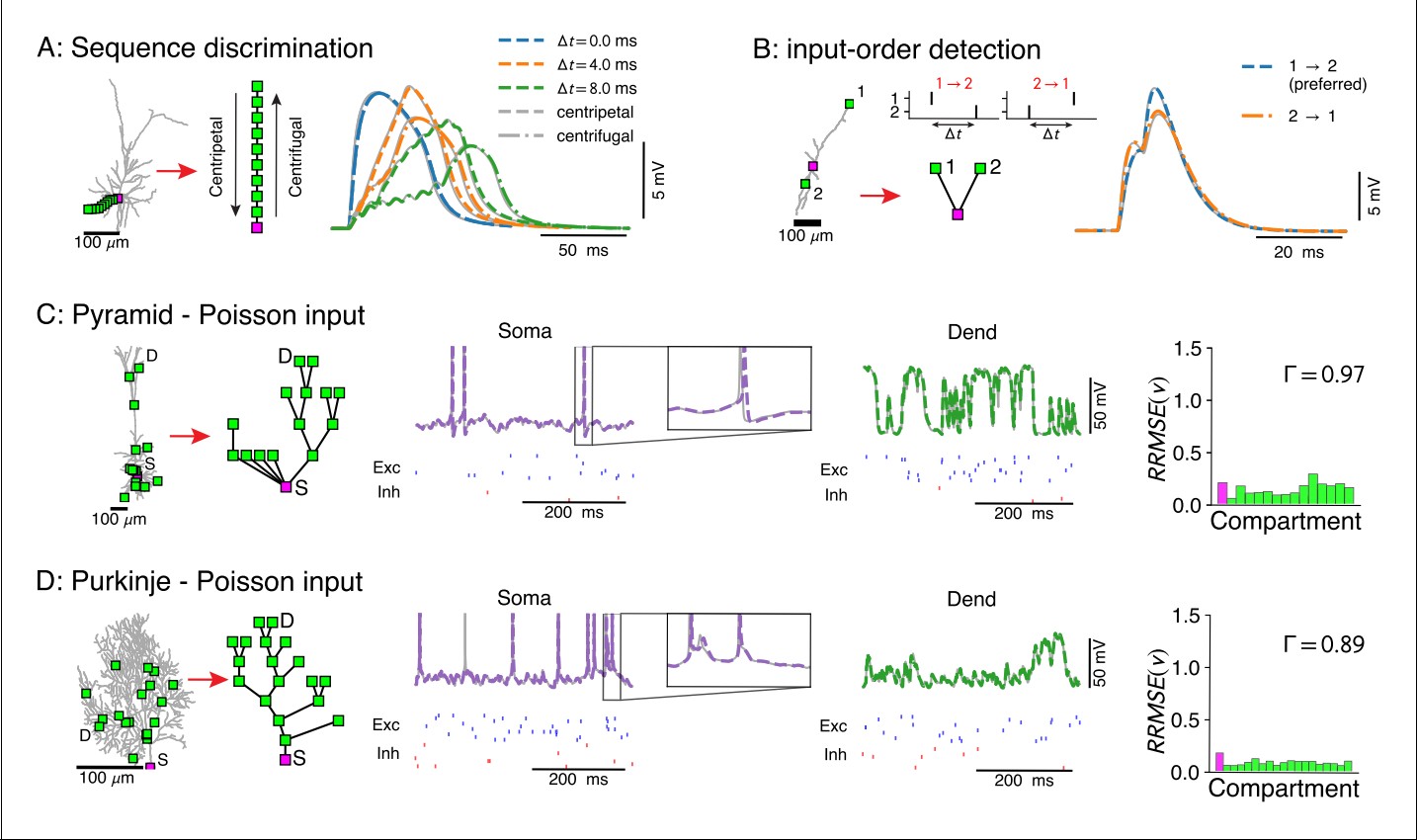

**Figure 2.** Voltage-match between full and reduced models for spatio-temporal dendritic computations. (**A**) Reduction of full model (right) to a single branch (middle) reproduces sequence discrimination (right), full model in gray, and reduced model colored for different time-steps between inputs, centripetal (dashed), and centrifugal (dash-dotted). (**B**) Full model (left) and three-compartment reduction (middle, bottom) discriminate temporal order of inputs, where the response to inputs (middle, top) ordered $1 \rightarrow 2$ is stronger than $2 \rightarrow 1$. Voltage responses (right) in full model (gray) and reduced model (colored). (**C, D**) Reductions of resp. L5 pyramidal and Purkinje cells with active ion channels at the soma, and excitatory (AMPA+NMDA) and inhibitory (GABA) synapses at dendritic compartment sites. From left to right: full model with compartment sites (soma $S$ and a selected dendrite site $D$ are labeled), reduced model, somatic voltage with zoom on a single AP (full model in gray and reduced model in purple, input spikes at the bottom), dendritic voltage at $D$ (full model in gray and reduced model in green, input spikes at the bottom), and the relative root mean square voltage errors at each compartment (root mean squared error normalized by the standard deviation $RRMSE(v) = \sqrt{\mathrm{Avg}(v_{\mathrm{full}} - v_{\mathrm{reduced}})^2} / \sigma_{v_{\mathrm{full}}}$). Spike coincidence factor $\Gamma$ is also shown.

reproduced the somatic and dendritic membrane voltage traces of their full counterparts (*Figure 2C,D*, middle panels). Furthermore, 97% and 89% of APs coincided within a 6 ms window, respectively (*Jolivet et al., 2008*; *Figure 2C,D*, right panels).

Models with passive dendritic membranes but non-linear synapse clusters can already reproduce much of the canonical computational repertoire of dendrites. Clusters of AMPA+NMDA synapses at the distal tips of basal dendrites can make neurons function as two-layer networks (*Mel, 1993*; *Poirazi et al., 2003b*; *Poirazi et al., 2003a*). Such reduced models also exhibit the same robustness to input noise as the full models (*Poleg-Polsky, 2019*), and can include modulation of the co-operativity between AMPA+NMDA clusters through compartments with shunting conductances (*Wybo et al., 2019*). Furthermore, because our reduced models reproduce dendritic input impedance properties, long time-scale NMDA currents activate strongly in distal compartments as high distal input impedance helps to overcome their voltage-dependent Mg$^{2+}$ block (*Jahr and Stevens, 1990a*; *Jahr and Stevens, 1990b*), whereas short time-scale AMPA currents dominate in more proximal compartments. Thus passive models with non-linear synapse clusters also capture the shift in computational strategy from a proximal temporal code to a distal rate code (*Branco and Häusser, 2011*).

What level of spatial complexity reproduces characteristic responses generated by somatic and dendritic ion channels? These responses arise through the interplay of integrative properties of the morphology – captured in the impedance kernels $z(t)$ – and channel dynamics. If a full model (here the L2/3 pyramid) is reduced to a single somatic compartment (*Figure 3A*), the impedance kernel has only one time scale – the membrane time-scale (*Figure 3B*, top – purple). The kernel of the full model contains additional shorter time scales (*Figure 3B*, top – gray), leading to a faster AP onset than in the reduced model (*Figure 3B*, bottom). This effect is thus a fundamental limitation of the

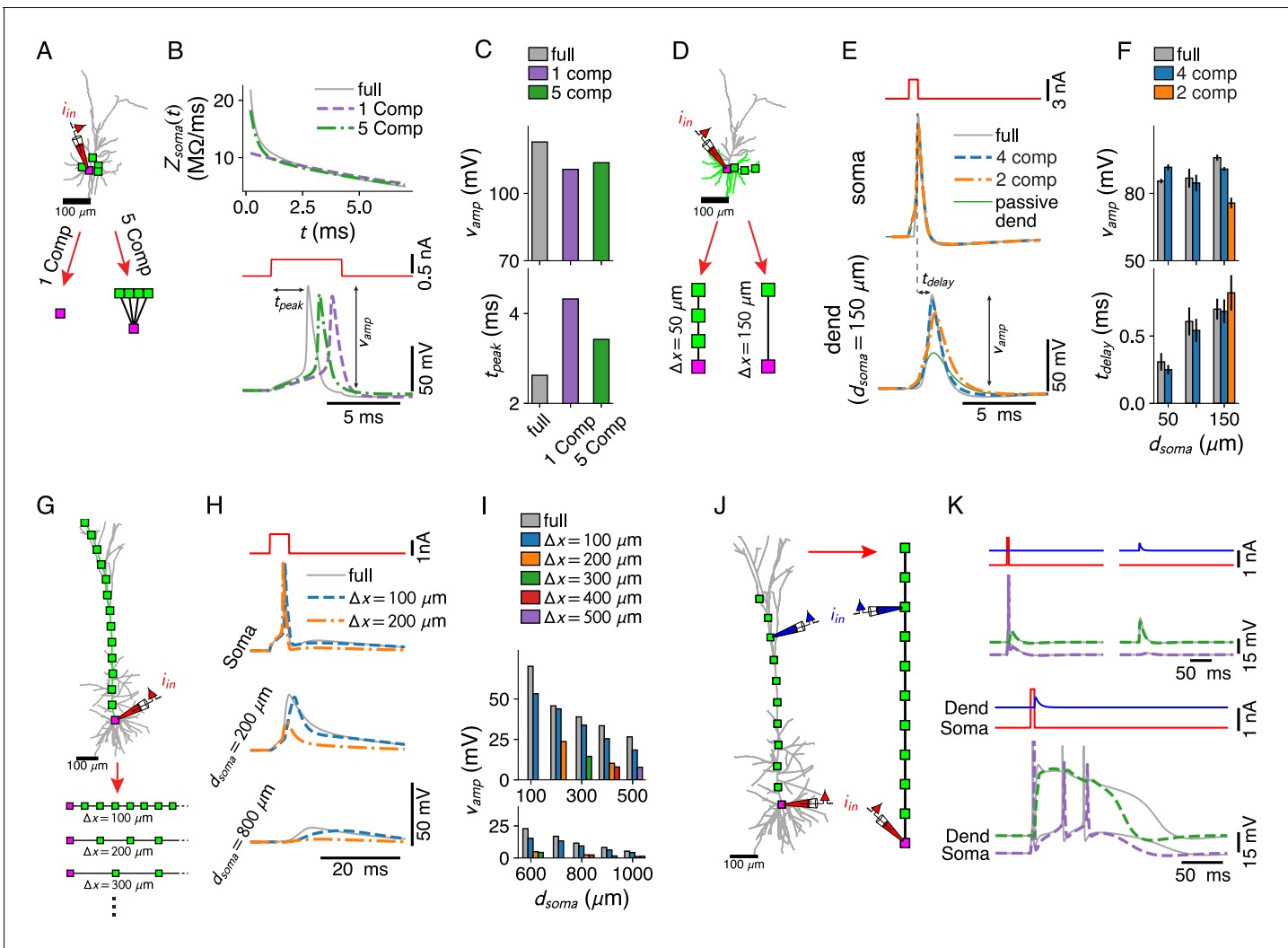

**Figure 3.** Dendritic computations with active channels are captured by our reduced models. (A–C) Effect of compartment distribution on AP dynamics in reduced models. (A) One- and five-compartment reductions of the L2/3 pyramid, equipped with somatic and dendritic ion channels. (B) Differences in short time-scale behavior in somatic input impedance kernels (top) between full model (gray) and one- (purple) and five-compartment (green) reductions result in different AP delays (bottom). (C) AP amplitude (top) and AP delay (bottom) for the three models (colors as in B). (D–F) Effect of compartment distribution on AP back-propagation in basal dendrites. (D) Four- and two-compartment reductions of a basal branch. (E) APs at soma (top) and most distal compartment site (bottom) for four models (full in gray, four compartments in blue, two compartments in orange, and full but with a passive dendrite in green). (F) Amplitude (top) and delay (bottom) for bAPs at different distances from soma (if compartment is present in model), averaged over all basal branches longer than 150 μm (error bars indicate standard deviation). (G–I) AP back-propagation in the apical dendrite of the L5 pyramid. (G) Reductions of the apical dendrite with increasing inter-compartment spacing. (H) Voltage waveform at soma (top, full in gray, $\Delta x = 100$ μm in blue, $\Delta x = 200$ μm in orange) and two dendritic sites (middle, bottom). (I) Waveform amplitude as a function of distance to soma for various inter-compartment spacings. (J, K) $Ca^{2+}$-spike-mediated coincidence detection. (J) Reduction of the L5-pyramid's apical dendrite to 11 compartments. (K) Response to a somatic current pulse (top, left), a dendritic synaptic current waveform (top, right), and the coincident arrival of both inputs (bottom). The online version of this article includes the following figure supplement(s) for figure 3:

**Figure supplement 1.** Further reduction of the configuration in *Figure 3J* by omitting compartments in the apical trunk of the L5 pyramid.

single-compartment reduction. Extending the compartmental model with four nearby dendritic sites (*Figure 3A*) adds fast components to the somatic impedance kernel (*Figure 3B*, top – green) that model the spread of charge to the other compartments. These components increase AP amplitude and decrease AP delay (*Figure 3C*), leading to a better match with the full model.

Dendritic ion channels support the back-propagation of APs (*Stuart et al., 1997*). For each basal branch of at least 150 μm in the L2/3 pyramid, we derived reduced models with three compartments (at 50, 100, and 150 μm from the soma) and with one compartment. We compared bAP amplitudes and delays relative to the somatic AP (*Figure 3E*) and found that even models with a single distal compartment support active back-propagation (*Figure 3E,F*). In the apical dendrite of the L5 pyramid (*Figure 3G*), we found that a distance step of 100 μm between compartments was required to support bAPs to the same degree as in the full model (*Figure 3H,I*).

Finally, we considered the pairing of a somatic current pulse with a dendritic post-synaptic potential waveform. We found that an 11-compartment reduction (*Figure 3J*) – with compartments spaced at 100 μm through the apical trunk to support bAPs until the $Ca^{2+}$ hot-zone – reproduced the $Ca^{2+}$-mediated coincidence detection mechanism (*Larkum et al., 1999*; *Pérez-Garci et al., 2013*).

## Conditions under which afferent spatial connectivity motifs can be simplified

Our method allows us to reduce morphological complexity by removing any branch or subtree without affecting the integrative properties of other branches or subtrees. Up until now, we have only considered reductions where the to-be removed branch or subtree does not receive direct synaptic input. However, if this branch or subtree does receive synaptic input, the morphological reduction also simplifies the afferent spatial connectivity motifs targeting it, as synapses have to be grouped at the nearest proximal intact compartment. We assess whether the computational repertoire of the neuron remains intact under such a simplification, or which elements of the repertoire might be lost. We study two proxies for this repertoire: the voltage difference between full and reduced models at the intact compartments – in order to assess whether the output of local computations in the ablated branch or subtree can be recovered, and the difference between full and reduced models in NMDA-spike threshold at the nearest proximal intact compartment (quantified as the smallest number of AMPA+NMDA synapses with a weight of 1 nS that need to be activated simultaneously to elicit an NMDA spike) – in order to assess whether local integrative properties at the intact compartments can also be retained under the shift of synapses. We allow ourselves the liberty of rescaling the weights of the shifted synapses by a temporally constant factor $\beta$ (*Figure 4A*). Which spatial synapse distributions admit simplification in this sense, and under which input conditions?

For a single, current-based synapse, shifted from its original location on the ablated branch to the next proximal compartment, we analytically compute the weight-rescale factor and find $\beta_{curr} = z_{cs}/z_{cc}$, with $z_{cs}$ the transfer resistance from synapse site $s$ to compartment site $c$ and $z_{cc}$ the input resistance at $c$. Since $z_{cs} < z_{cc}$ (*Koch, 1998*), the weight-rescale factor weakens the shifted synapse, so that it matches the attenuation of its distal counterpart. Nevertheless, when $s$ is located more distally on the same branch or subtree as $c$, the transfer resistance $z_{cs}$ is often close to the input resistance $z_{cc}$ (*Figure 1—figure supplement 1E*). Thus, $\beta_{curr}$ is often close to one, and hence negligible (*Figure 4B,C*). For a conductance-based synapse, rescaling weights by $\beta_{curr}$ does not accurately fit the full model (*Figure 4B*). We decompose the synaptic conductance $g(t)$ as $g + \delta g(t)$, with $g$ the temporal average and $\delta g(t)$ the fluctuations around $g$. We then analytically compute (see Materials and methods — Synaptic weight-rescale factors) that the weight-rescale factor for a conductance-based synapse

$$\beta_{cond} = \frac{1}{1 + \Delta z\, g} \tag{8}$$

recovers the full model's voltage if

$$\Delta z\, \delta g(t) \ll 1 + \Delta z\, g, \tag{9}$$

where $\Delta z = z_{ss} - z_{cc}$ is the difference between input resistance $z_{ss}$ at the original synapse site $s$ and input resistance $z_{cc}$ at the nearest proximal compartment site $c$. Multiplying the synaptic weights by $\beta_{cond}$ weakens the synapse so that it matches the shunt effect of its distal counterpart, and so that

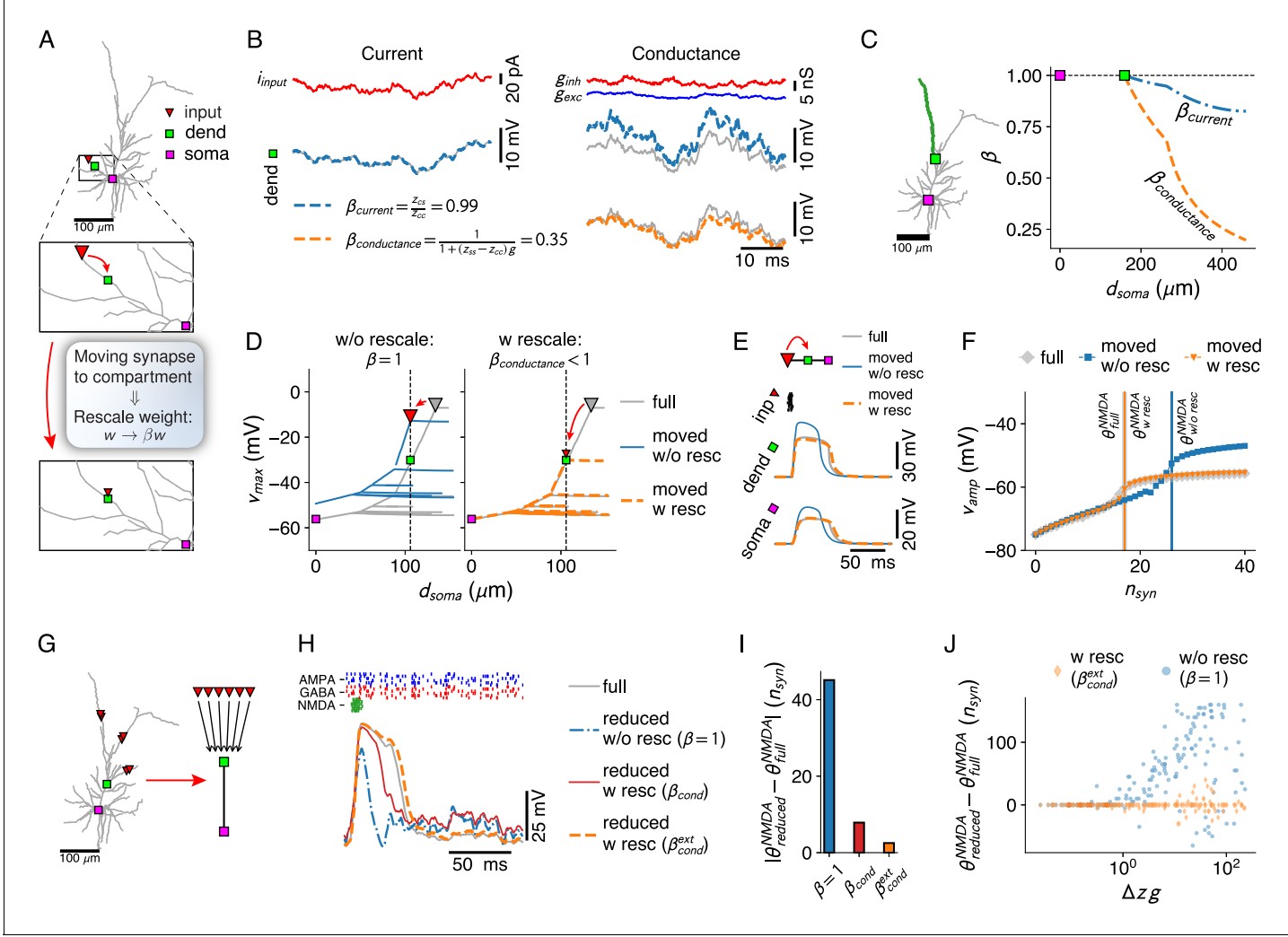

**Figure 4.** Simplification of afferent spatial connectivity motifs. (**A**) Removal of a branch with a synapse (red triangle) is considered possible if the correct voltages at the compartment sites (here, dendritic – green square and somatic – purple square) can be obtained by shifting the synapse to the compartment site and rescaling its weight with a fixed factor $\beta$ for the all input conditions under consideration. (**B**) Comparison between weight-rescale factors for current-based (left) and conductance-based (right) input. Voltage trace for dendritic compartment without rescaling (gray) and with the current-based scale factor $\beta_{curr}$ compensating for attenuation (blue). Bottom right panel shows voltage trace with conductance-based scale factor $\beta_{cond}$. (**C**) Current- and conductance-based scale factors in the green dendritic branch, for a shift of a synapse at a given distance from the soma to the dendritic compartment site (green square). (**D**) Spatial peak voltage without (left, blue) and with (right, orange) application of $\beta_{cond}$. (**E**) For a cluster of AMPA+NMDA synapses in isolation, scale factors for physiological constants can be obtained (see Materials and methods — Synaptic weight-rescale factors) that reproduce the correct voltage waveform. Colors as in D. (**F**) Maximal amplitude of NMDA-spike waveform upon activation of increasing numbers of synapses – NMDA-spike thresholds indicated with vertical lines. Colors as in D. (**G–H**) Removing a whole subtree and shifting multiple synapses (red triangles) to the next proximal compartment site (green square). NMDA-spike generation (gray voltage trace) at the compartment site through burst inputs to local AMPA+NMDA synapses (green inputs), with AMPA (blue) and GABA (red) background inputs spread throughout the subtree. Reductions shown without rescaling (blue, dash-dotted), with the analytical single-site rescaling rule (red, full) and the numerical multi-site rule (orange, dashed). (**I**) Error in NMDA-spike threshold for the three cases in H. (**J**) Dependence of the error in NMDA-spike threshold on the factor $\Delta z\,g$, with $\Delta z$ the average input resistance difference between synapse sites and the compartment and $g$ the average synaptic conductance.

the voltage at all compartments more proximal than $c$ and in their respective side branches is reproduced (***Figure 4D***).

If $s$ and $c$ are close, $\Delta z$ is small and (***Equation 9***) is satisfied for plausible conductance fluctuations. Thus, shifting the synapse to the next proximal compartment keeps the voltage response intact. If the separation between $s$ and $c$ increases, $\Delta z$ increases accordingly, and the magnitude of tolerated conductance fluctuations shrinks (***Equation 9***). Thus, for a given magnitude of $\delta g(t)$, it is possible

that a thick branch can be removed, but not a thin branch, as the input resistance increase in the former is much more gradual than in the latter. Nevertheless, (*Equation 9*) suggests that even for large $\Delta z$ the full model's voltage can be recovered by downscaling synaptic weight by $\beta_{\text{cond}}$, but only if $\delta g(t) \ll g$. A tonic level of AMPA and/or GABA activation, as in the high-conductance state (*Destexhe et al., 2003*), can implement this input condition. For NMDA synapses, this input condition does not hold, as NMDA spikes arise through a strong, transient conductance increase. However, when a cluster of such synapses – to be moved to a proximal site – is considered in isolation, a workaround can be found by applying constant rescale factors not just to synaptic weight, but also to threshold and width of the $Mg^{2+}$ block, as well as to the reversal (see Materials and methods — Synaptic weight-rescale factors). These factors recover NMDA-spike shape (*Figure 4E*) and threshold (*Figure 4F*) when the cluster is moved to from $s$ to $c$.

Our analytical weight-rescale factor $\beta_{\text{cond}}$ treated reductions where a single site $s$ was moved to $c$. We consider now the ablation of a whole subtree and the shift of all its synapses to $c$ (*Figure 4G*). In our simulations, we distributed 100 AMPA and 100 GABA synapses on the to be ablated subtree, and activated them with a Poisson process with fixed rate. We studied how the reduction influenced the local integrative properties at the intact compartment $c$. As a proxy for those integrative properties, we checked the shape of the waveform elicited by activating AMPA+NMDA synapses at $c$ in a short burst (*Figure 4H*), and we quantified the threshold for NMDA-spike generation. We found that neither shape nor threshold were preserved by the analytical weight-rescale factors $\beta_{\text{cond}}$, derived for the case where only a single input site is present on the subtree. We numerically extend the derivation of the weight-rescale factors to multi-site inputs (see Materials and methods – Synaptic weight-rescale factors). These extended weight-rescale factors $\beta_{\text{cond}}^{\text{ext}}$ reproduced local voltage (*Figure 4H*) and NMDA-spike threshold (*Figure 4I*). We also investigate whether $\beta_{\text{cond}}^{\text{ext}}$ can still be conceptualized as depending on $\Delta z_{\text{avg}} g_{\text{avg}}$ – the product between (1) a difference in average input resistance between synapses on the to be ablated subtree and the compartment site and (2) the average conductance load exerted by those synapses. To do so, we activated the 100 AMPA and 100 GABA synapses on the to be ablated subtree at a wide range of input conditions (see Materials and methods – Simulation-specific parameters), and measured the error in threshold for NMDA-spike generation at $c$. We found that this error depended strongly on $\Delta z_{\text{avg}} g_{\text{avg}}$ and remained small for $\Delta z_{\text{avg}} g_{\text{avg}} \ll 1$ (*Figure 4J*).

In conclusion, we can now assess whether given spatial synapse motifs on branches or subtrees admit simplification by shifting the synapses to the nearest proximal compartment site and ablating the branch or subtree. We find that reduction in this sense is possible for any motif if fluctuations are small, so that $\delta g(t) \ll g$, by applying temporally constant weight-rescale factors. Otherwise, the synapses must be located sufficiently closely to the compartment site, so that $\Delta z$ is small and that $\Delta z \, \delta g$ remains below $1 + \Delta z \, g$ for all relevant activation levels.

## Active and passive reduced dendrites under synaptic bombardment

We next investigated how many compartments are needed to reproduce somatic and dendritic responses under a synaptic bombardment and quantified the contribution of active dendritic $Na^+$, $Ca^{2+}$, and $K^+$ channels. We mimicked the in vivo state in the basal dendrites of the L2/3 pyramid by distributing 200 AMPA and 200 GABA synapses receiving Poisson inputs and 20 clusters of AMPA +NMDA synapses receiving bursts of inputs (*Figure 5A*). We sampled 10 different sets of meta-parameters governing synaptic activation (Poisson and burst rates and synaptic weights, see Materials and methods — Simulation-specific parameters), resulting in output spike rates between 0 and 8 Hz (*Figure 5B*). We derived reduced models, once based on the full model with all active channels ('active reduced dendrite' in *Figure 5C*) and once based on the passified full model ('passive reduced dendrite' in *Figure 5C*). We distributed increasing numbers of compartment sites on the basal branches and measured somatic and dendritic voltage responses for the active full model and for the reduced models with active and passive dendrites (*Figure 5C*, *Figure 5—figure supplement 1* for dendritic traces). Spike coincidence (*Figure 5D*), subthreshold somatic voltage error (*Figure 5E*), and dendritic voltage error (*Figure 5F*) showed a significant improvement when compartment numbers increased from 0 (point-neuron – top row in *Figure 5C*) to 2 per 100 µm. In the presence of active channels, the error decreases until up to three compartments per 100 µm

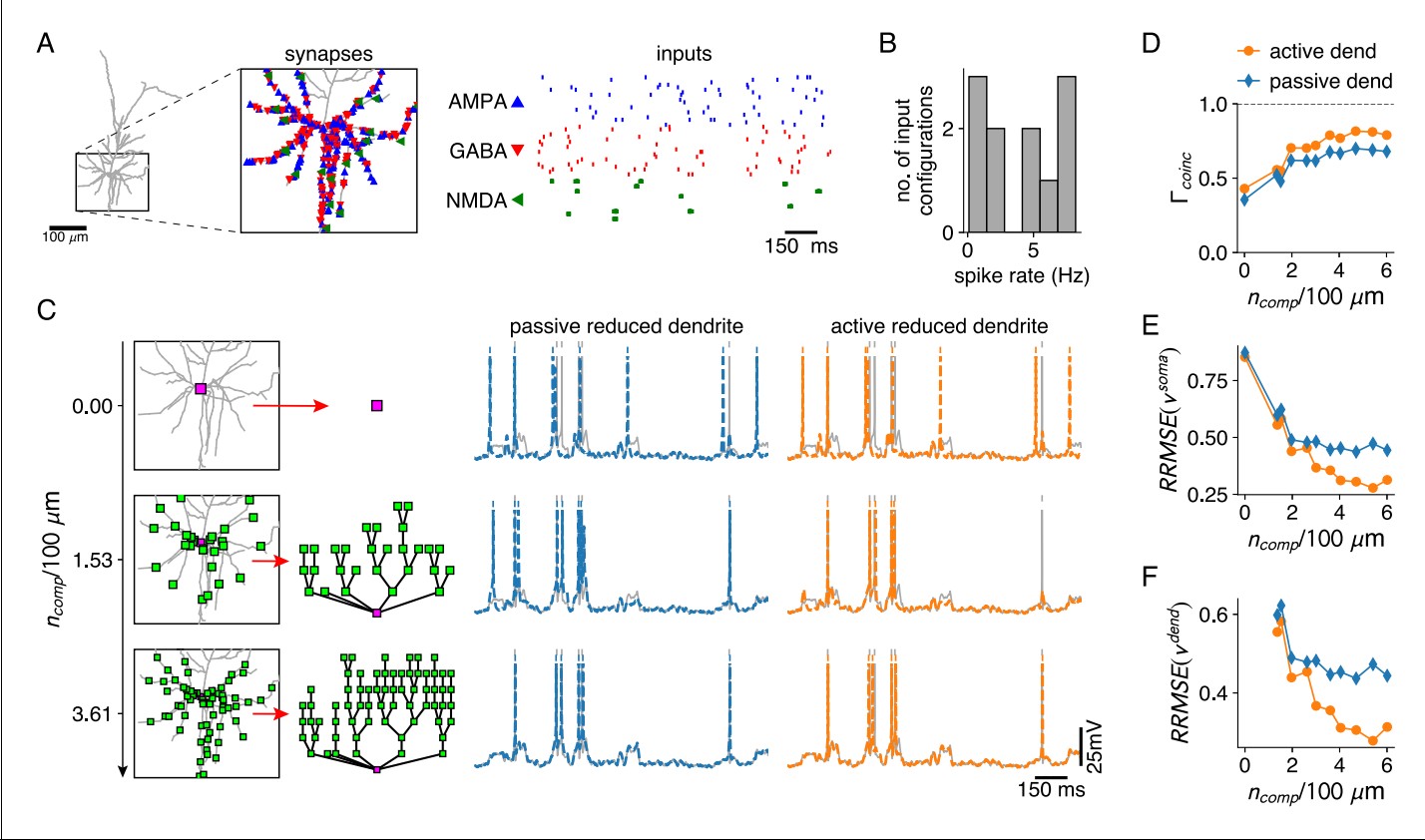

**Figure 5.** Reductions with active and passive dendritic compartments under in vivo like conditions. (**A**) AMPA and GABA synapses, and AMPA+NMDA synapse clusters are spread randomly throughout the basal dendrites of the L2/3 pyramid (inset). AMPA and GABA synapses receive Poisson inputs and AMPA+NMDA synapses receive bursting input (right). (**B**) Output spike rates of the full model for 10 different input configurations. (**C**) Reductions with increasing numbers of compartment sites along the basal dendrites (left). Somatic voltage traces and spike times for the full model (gray), for a reduction with passive dendrites (middle, blue) and a reduction with active dendrites (right, orange). (**D–F**) Spike coincidence factors (**D**), relative somatic voltage errors (**E**) and relative dendritic voltage errors (**F**) for reductions with increasing numbers of compartments, and with passive (blue) and active dendrites (orange), averaged over all 10 input configurations. Relative voltage errors are computed as in *Figure 2C,D*.

The online version of this article includes the following figure supplement(s) for figure 5:

**Figure supplement 1.** Dendritic voltage traces for reductions with passive and active dendrites.

(*Figure 5E,F*). Spike coincidence factors were consistently ~10% higher than in the passive case (*Figure 5D*).

## Fitting reduced models directly to experimental data

Traditionally, creating a morphological neuron model involves a reconstruction of the morphology and an evolutionary fit of the electrical parameters (*Hay et al., 2011*; *Almog and Korngreen, 2014*; *Van Geit et al., 2016*). Our method allows skipping these resource-intensive steps (*Figure 6A*). We adapt our method to a common experimental paradigm where hyper- and depolarizing current steps are injected under applications of various ion-channel blockers (*Marti Mengual et al., 2020*). We extract resistance matrices and holding potentials from voltage step amplitudes (*Figure 6B*), and the time scale of the largest eigenmode from the average voltage decay back to rest (*Figure 6C*). If the modeling goal is a reduced model, the traditional reconstruction and subsequent reduction both introduce errors (*Figure 6D*). Thus, we find that while our reduction of the full model is faithful within an expected accuracy margin (*Figure 6E*), fitting a reduced model directly to the experimental traces is more accurate (*Figure 6F*).

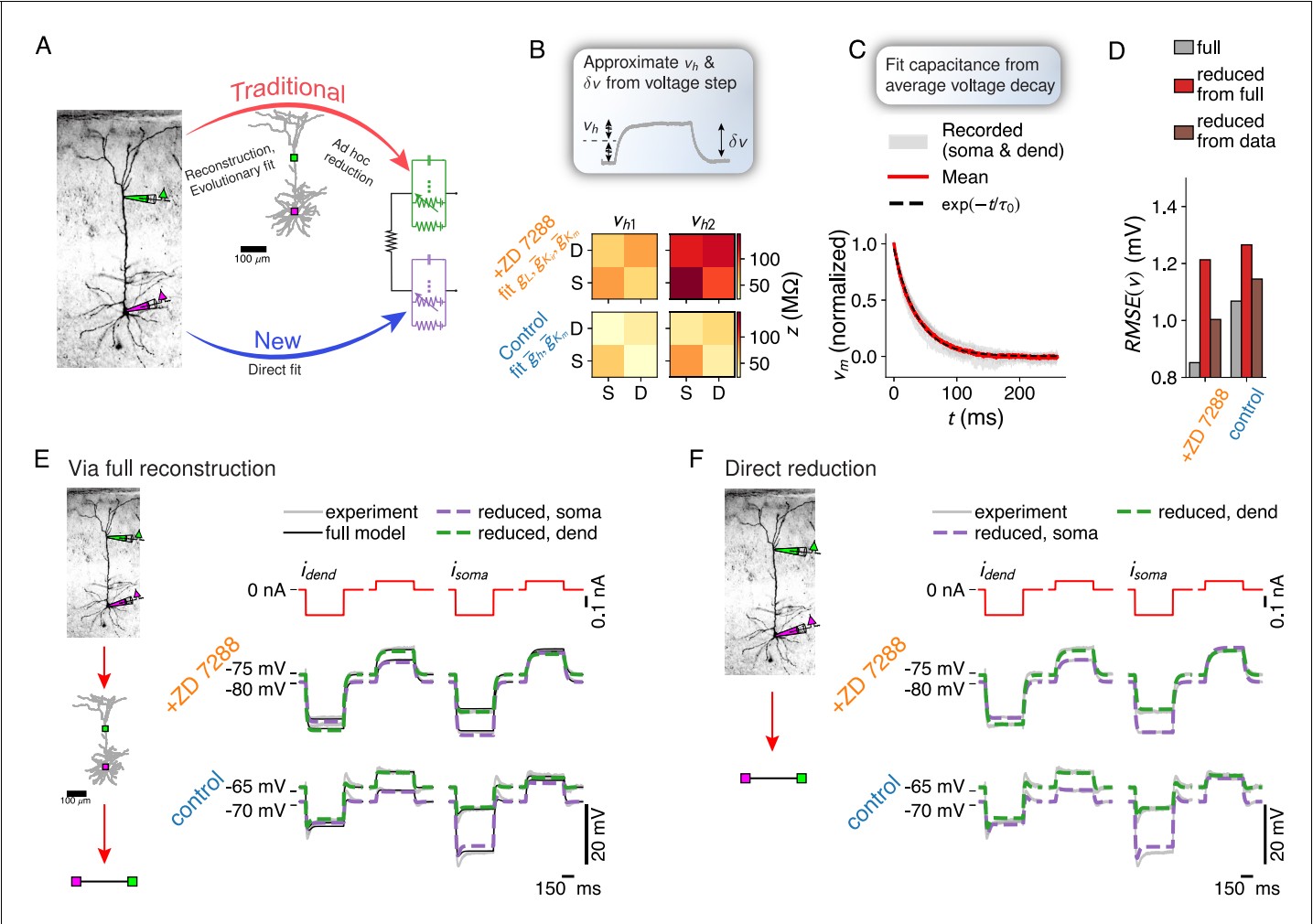

**Figure 6.** Fitting reduced models directly to experimental data. (**A**) Comparison between the traditional neuron model creation paradigm (morphological reconstruction and evolutionary fit, possibly followed by a reduction), and the proposed direct experiment to reduced dendrite model paradigm. (**B**) Resistance matrices and holding potentials are extracted from the response amplitude to hyper- and depolarizing current step inputs, here measured once under application of an h-channel antagonist and once under control conditions. (**C**) Time scale of largest eigenmode is extracted from average decay back to the resting membrane potential. (**D**) Combined root mean square voltage error $RMSE(v) = \sqrt{\mathrm{Avg}(v_{\mathrm{full}} - v_{\mathrm{reduced}})^2}$ of full model (gray) – fitted to current step data through an evolutionary algorithm, of the reduction of the full model (red), and of the direct fit of the reduced model to the data (brown). (**E**) Experimentally recorded traces (gray), traces from the full reconstruction (black), and its reduction (green and purple, dashed). (**F**) Experimentally recorded traces (gray) and traces from the directly fitted reduced model (green and purple, dashed).

## Discussion

We have presented a flexible yet accurate method to construct reduced neuron models from experimental data and morphological reconstructions. The method consists of linear parameter fits at holding potentials that are informative for the full non-linear dynamics. First, we derive leak and coupling conductances from the passified version of the full model. Second, we fit the maximal conductances of the active ion-channels, so that the reduced model, at various holding potentials, responds similarly to input perturbations as the full model. Third, we fit the capacitances to reproduce spatio-temporal integrative properties. Finally, we ensure that resting membrane potentials in the full and reduced models match. The resulting method can be adapted to a wide variety of use cases.

A fundamental use case is deriving reduced models that retain specific elements of the dendritic computational repertoire, to explore how those dendritic computations may improve network function. To that purpose, one can apply our method to compartments placed at input sites needed for

the computation of interest. For instance, if a computation requires independent computational sub-units (*Poirazi et al., 2003b*; *Kastellakis et al., 2016*), one would place compartments at the dendritic tips. If one wants to modulate co-operativity of excitatory inputs at distal sites, one would place compartments with shunting conductances at the bifurcations between the distal sites (*Wybo et al., 2019*). Detecting input sequences (*Branco and Häusser, 2010*) or implementing a dendritic integration gradient (*Branco and Häusser, 2011*) would require placing two or more compartments along the branches of interest. For many of these computations, a passive morphology with non-linear (NMDA) synapses suffices. Furthermore, our method can be combined with abstract models of AP generation (*Pozzorini et al., 2013*; *Pozzorini et al., 2015*).

A second use case is constructing models directly from electro-physiological recordings. Our method avoids the labor-intensive detour of reconstructing the morphology and optimizing model parameters with an evolutionary algorithm (*Marti Mengual et al., 2020*). In combination with advances in voltage-sensitive dye imaging, our method may thus form the basis of a high-throughput experiment-to-model paradigm.

A third use case is elucidating the effective complexity of a given dendritic tree. We assessed whether afferent spatial input motifs could be simplified by removing a branch or subtree and grouping the affected synapses at the next proximal compartment. We found that such simplification is possible when the input resistance of the affected synapse sites is close to the input resistance of the compartment site. If this condition does not hold, the simplification can still proceed if conductance fluctuations are small compared to the average conductance magnitude, but not otherwise. What is the simplest model for a given dendritic tree that captures its full computational repertoire? We find that spike coincidence and subthreshold voltage metrics reach satisfying accuracy at, but do not improve much beyond ~3 compartments per 100 μm. Further research is required to tease apart mere numerical errors from potentially missed computational features.

The process of simplifying complex dynamical systems, such as morphological neuron models, is inherently approximate. Due to the vast variety of morphologies, voltage-gated ion-channel configurations and input patterns, it is beyond the scope of any single paper to give an exhaustive overview of possible inaccuracies. Rather, we propose three lines of inquiry if a reduction fails to reproduce a particular response pattern. First, we suggest checking the discrepancy between the impedance kernels $z(t)$ of the full and reduced models. Often, response mismatches arise from slight inaccuracies in how ion-channel dynamics are integrated by the passive system (*Figure 3A–C*). Second, we suggest assessing to what degree afferent spatial input motifs, present in the full model, are simplified, and whether that simplification still admits a sufficient accuracy in reproducing local voltage-dependent responses, such as NMDA spikes (*Figure 4*). Finally, we suggest checking whether the distribution of compartments on the morphology admits ion channel dynamics that are sufficiently rich to reproduce the response characteristic in question. To illustrate these lines of inquiry, we have further simplified the L5 pyramid to five compartments, by omitting the compartments in the apical trunk (*Figure 3—figure supplement 1A* vs *Figure 3J*). This reduction results a change in AP shape, and in a failure to elicit AP bursts through the pairing of a somatic and a dendritic input, even though a $Ca^{2+}$ spike can still be elicited (*Figure 3—figure supplement 1B*). Likely, the change in AP shape can be traced back to the difference in the somatic input impedance kernels (*Figure 3—figure supplement 1C*), while the failure to elicit an AP burst may be due to insufficient $Na^+$-driven cross-talk between soma and apical tuft.

Computational tools have played a key role in neuroscience. Take NEURON (*Carnevale and Hines, 2004*), which accelerated our understanding of morphological neurons, or NEST (*Gewaltig and Diesmann, 2007*), which enabled simulation of large-scale spiking point-neuron networks. We have implemented a Python toolbox that automatizes the simplification process (NEural Analysis Toolkit – NEAT – https://github.com/unibe-cns/NEAT; copy archived at swh:1:rev: 1cb15f36aa0a764105348541d046c85ef38e21ee). The toolbox reads any morphology in the standard '.swc' format (*Ascoli, 2006*) and returns the parameters of the reduced models, while also providing tools to export the models to NEURON (*Carnevale and Hines, 2004*).

Our method and toolbox fill a void in between the extremes of modeling large-scale networks with abstract models and modeling single cells in all their detail. By enabling the efficient systematic derivation of simplified neurons, amenable to simulation at the network level, our work bridges the gap between two branches of neuroscience that historically have remained separate.

## Materials and methods

### Morpohologies

Three exemplar cell models were used for the analysis: a cortical L2/3 pyramidal cell (*Branco and Häusser, 2010*; *Figure 2D*), a cortical L5 pyramidal cell (*Hay et al., 2011*; *Figure 2E*), and a cerebellar Purkinje cell (*Figure 2F*). The latter morphology was retrieved from the NeuroMorpho.org repository (*Ascoli, 2006*) and the two others from the ModelDB repository (*McDougal et al., 2017*).

### Physiological parameters

In our passive models, physiological parameters were set according to *Major et al., 2008*: the equilibrium potential was $-75\,\mathrm{mV}$, the membrane conductance $100\,\mu\mathrm{S/cm}^2$, the capacitance $0.8\,\mu\mathrm{F/cm}^2$, and the intracellular resistivity $100\,\Omega\cdot\mathrm{cm}$. For the active models, we took ion channels and parameters according to *Branco and Häusser, 2010* for the L2/3 pyramid, *Hay et al., 2011* for the L5 pyramid, and *Miyasho et al., 2001* for the Purkinje cell.

AMPA and GABA synaptic input currents were implemented as the product of a conductance profile, here a double exponential (*Rotter and Diesmann, 1999*), with a driving force:

$$i_{\mathrm{syn}} = g\,(e_r - v). \tag{10}$$

AMPA rise resp. decay times were $\tau_r = 0.2\,\mathrm{ms}$, $\tau_d = 3\,\mathrm{ms}$ and AMPA reversal potential was $e = 0\,\mathrm{mV}$. For GABA, we used $\tau_r = 0.2\,\mathrm{ms}$, $\tau_d = 10\,\mathrm{ms}$, and $e = -80\,\mathrm{mV}$. NMDA currents (*Jahr and Stevens, 1990a*) had the form:

$$i_{\mathrm{syn}} = g\,\sigma(v)\,(e - v) \tag{11}$$

with rise resp. decay time $\tau_r = 0.2\,\mathrm{ms}$, $\tau_d = 43\,\mathrm{ms}$, and $e = 0\,\mathrm{mV}$, while $\sigma(v)$, modeling the channel's magnesium block, had the form (*Behabadi and Mel, 2014*):

$$\sigma(v) = \frac{1}{1 + 0.3\,e^{-0.1\,v}}. \tag{12}$$

The 'conductance' of an AMPA or GABA synapse signifies the maximum value of its conductance window. For an AMPA+NMDA synapse, the conductance is the maximal value of the AMPA conductance window, and the conductance of the NMDA component is determined by multiplying the AMPA conductance value with an NMDA ratio $R_{\mathrm{NMDA}}$, set to be either 2 or 3.

### Biophysical models

We used the NEURON simulator (*Carnevale and Hines, 2004*) to implement biophysical and reduced models. For the biophysical models, the distance step was set according the lambda rule (*Carnevale and Hines, 2004*) or smaller.

### Quasi-active channels

A voltage-dependent ion channel described by the Hodgkin–Huxley formalism can in general be written as follows:

$$i = \overline{g}f(y_1,\ldots,y_K)\,(v - e), \quad \dot{y}_k = g_k(y_k,v) \text{ for } k = 1,\ldots,K, \tag{13}$$

where $\overline{g}$ is the channel's maximal conductance, $e$ its reversal potential, $y_1,\ldots,y_K$ its state variables, $f(\blacksquare)$ a function that depends on the channel type (e.g. for a typical sodium channel $f(m,h) = m^3 h$), and $g_k(y_k,v)$ $(k = 1,\ldots,K)$ the functions governing state-variable activation ($g_k(y_k,v)$ can usually be written as $(y_{\infty k}(v) - y_k)/\tau_k(v)$ with $y_{\infty k}(v)$ the state-variable's activation and $\tau_k(v)$ its time scale). To obtain the channel's quasi-active approximation (*Mauro et al., 1970*; *Koch and Poggio, 1985*) around a holding potential $v_h$ and a state variable expansion point $y_1^0,\ldots,y_K^0$, we linearize (*Equation 13*):

$$i_{\text{lin}} = \overline{g}\left[\sum_{k=1}^{K}\frac{\partial f}{\partial y_k}(y_1^0,\ldots,y_K^0)\,(v_h - e)\,(y_k - y_k^0) + f(y_1^0,\ldots,y_K^0)\,(v - v_h)\right],$$

$$\dot{y}_k = \frac{\partial g_k}{\partial y_k}(y_k^0, v_h)\,(y_k - y_k^0) + \frac{\partial g_k}{\partial v}(y_k^0, v_h)\,(v - v_h) \quad \text{for } k = 1,\ldots,K. \tag{14}$$

To obtain the zero-frequency contribution of (*Equation 14*) to the resistance and conductance matrices, we set $\dot{y}_k = 0$, solve the linearized state-variable equations for $(y_k - y_k^0)$, and substitute the result in $i_{\text{lin}}$:

$$i_{\text{lin}} = \overline{g}\left[-\sum_{k=1}^{K}\frac{\frac{\partial f}{\partial y_k}\frac{\partial g_k}{\partial v}}{\frac{\partial g_k}{\partial y_k}}(v_h - e) + f(y_1^0,\ldots,y_K^0)\right](v - v_h). \tag{15}$$

Introducing a shorthand for the factor in square brackets

$$l(v_h) = -\sum_{k=1}^{K}\frac{\frac{\partial f}{\partial y_k}\frac{\partial g_k}{\partial v}}{\frac{\partial g_k}{\partial y_k}}(v_h - e) + f(y_1^0,\ldots,y_K^0) \tag{16}$$

results in the linearized ion-channel current as described in (*Equation 2*).

## The impedance matrix

The voltage response $\delta v_x(t)$ at a location $x$ along the neuron to an input current perturbation $\delta i_{x'}(t)$ at location $x'$ can be computed as the convolution of an impedance kernel with $\delta i_{x'}(t)$:

$$\delta v_x(t) = z_{xx'}(t) \star \delta i_{x'}(t). \tag{17}$$

The impedance kernel itself can be computed in the frequency domain from the quasi-active cable equation using Koch's algorithm (*Koch and Poggio, 1985*). We may assume any a priori arbitrary set of holding potentials and ion-channel state variables to compute the quasi-active expansion (with Koch's algorithm, the only constraint is that their distribution is uniform for each cylindrical segment). For our purpose, spatially uniform holding potentials and state-variable expansion points suffice. For a steady-state current $\delta i_{x'}$, (*Equation 17*) simplifies to:

$$\delta v_x = z_{xx'}\,\delta i_{x'}, \tag{18}$$

where we term $z_{xx'} = \int_0^\infty \mathrm{d}t\, z_{xx'}(t)$ the 'resistance' ($z_{xx'}$ is also known as the input resistance if $x = x'$ or the transfer resistance if $x \neq x'$). For passive membranes, impedance kernels can also be computed as a sum of exponentials (*Holmes et al., 1992*; *Major et al., 1993*):

$$z_{xx'}(t) = \sum_{l=0}^{\infty}\phi_l(x)\,\phi_l(x')\,e^{-\frac{t}{\tau_l}}, \tag{19}$$

where we adopt the ordering $\tau_0 \geq \tau_1 \geq \tau_2 \geq \ldots$ Essentially, this infinite sum can be thought of as the generalization of the eigenmodes for linear dynamical systems (*Strogatz, 2000*) to partial differential equations.

When a current perturbation is applied at multiple sites along the neuron (we write $\delta\mathbf{i}(t) = (\delta i_1(t),\ldots,\delta i_n(t))$), we obtain the voltage response $\delta\mathbf{v}(t) = (\delta v_1(t),\ldots,\delta v_n(t)$ at those sites from:

$$\delta\mathbf{v}(t) = Z(t) \star \delta\mathbf{i}(t), \tag{20}$$

with $Z(t)$ the matrix of impedance kernels (the $ij$'th elements of this matrix is the impedance kernel between sites $i$ and $j$). In the steady-state case, we call $Z$ the resistance matrix.

## Compartmental models

The voltage $v_i$ in a compartment $i$, connected to a set $\mathcal{N}_i$ of nearest neighbor compartments, and with a set of ion channels $\mathcal{I}_i$, is given by

$$c_i \dot{v}_i + g_{Li}(v_i - e_{Li}) + \sum_{n \in \mathcal{N}_i} g_{C,in}(v_i - v_n) + \sum_{d \in \mathcal{I}_i} i_{di} = i_i, \tag{21}$$

where $c_i$ is the capacitance of compartment $i$, $g_{Li}$ resp. $e_{Li}$ its leak conductance resp. reversal, $g_{Cin}$ its coupling conductance to the neighboring compartment $n$, $i_{di}$ the ion channel current (*Equation 13*) of channel $d$ in compartment $i$ and $i_i$ an arbitrary input current to compartment $i$.

## The conductance matrix

To obtain the conductance matrix of the compartmental model for steady-state inputs, (*Equation 21*) is linearized around a certain holding potential $v_h$ and we assume that $\dot{v}_i = 0$. After absorbing all constant terms into $i_i$, we obtain

$$g_{Li}\delta v_i + \sum_{n \in \mathcal{N}_i} g_{C,in}(\delta v_i - \delta v_n) + \sum_{d \in \mathcal{I}_i} \overline{g}_{di} l_d(v_h) \delta v_i = i_i, \tag{22}$$

where $\delta v_i = v_i - v_h$, where $\overline{g}_{di} l_d(v_h)$ is the linearized ion-channel current (*Equation 15*), and where we used the notation (*Equation 16*). Summarizing the voltage responses for each compartment in a vector $\delta \mathbf{v} = (\delta v_1, \ldots, \delta v_N)$, (*Equation 22*) for each compartment $i = 1, \ldots, N$ can be recast as a matrix equation, yielding (*Equation 3*).

## Simplification method details

For any given set of $M$ sites on the morphology, we construct a simplified compartmental model whose connection structure follows a tree graph defined by the original morphology (*Figure 1A*). We do not allow triplet connections (e.g. sites 1–2, sites 2–3, and sites 3–1 all mutually connected) in our reduced models. Hence, to obtain accurate results, it is necessary to extend the original set of $M$ sites with the $B$ bifurcation points that lie in between (*Figure 1—figure supplement 1C*). With these $N = M + B$ sites, we thus define a tree graph that provides the scaffold for our fit (and our reduced model).

The fitting process proceeds in four steps: (1) fit the passive leak and coupling conductances, (2) fit the capacitances, (3) fit the maximal conductances of the ion channels, and (4) fit the reversal potentials to obtain the same resting membrane potentials as in the biophysical model. The order of steps 2 and 3 is interchangeable, but not of the other steps. We describe these steps in detail below:

1. If the biophysical model contains active conductances, we compute their opening probabilities at rest and add them to the leak. Otherwise, we simply take the leak as is. We then compute $Z$ for the $N$ compartment sites. In the passive case, $Z$ is independent of the holding potential. In (*Equation 21*), we substitute $\dot{v}_i = 0$, $i_c = 0$. The terms $g_{Li} e_{Li}$ are a constant contribution, and can thus be absorbed in $i_i$. We obtain:

$$g_{Li} v_i + \sum_{n \in \mathcal{N}_i} g_{C,in}(v_i - v_n) = i_i, \quad i = 1, \ldots, n \tag{23}$$

   which, when recast in matrix form, yields (*Equation 3*). (*Equation 4*) then yields a system of $N^2$ equations, linear in $g_{Li}$ and $g_{C,in}$ (note that this system always has $2N - 1$ unknowns). We recast this system of equations in a form $A\mathbf{g} = \mathbf{b}$ and solve it in the least-squares sense for $\mathbf{g}$.

2. To fit the capacitances, we set $i_c = 0$ and again absorb $g_{Li} e_{Li}$ in $i_i$ in (*Equation 21*) to obtain:

$$\dot{v}_i = \frac{1}{c_i}\left(-g_{Li} v_i + \sum_{n \in \mathcal{N}_i} g_{C,in}(v_n - v_i)\right) + \frac{i_i}{c_i}, \quad i = 1, \ldots, n. \tag{24}$$

   In matrix form, we obtain (*Equation 6*) with $S = \text{diag}(\mathbf{c})^{-1} G$. We require that this matrix has the same smallest eigenvalue (corresponding to the largest time scale $\tau_0$) $\alpha_0 = 1/\tau_0$ and eigenvector $\phi_0 = (\phi_0(x_1), \ldots, \phi_0(x_N))$ as the biophysical model, as defined by (*Equation 19*). Hence, we obtain the system of (*Equation 7*) that can be solved for $\mathbf{c}$.

3. We compute the maximal conductances of each ion-channel type separately. Thus, next to setting $\dot{v}_i = 0$, we set $i_c = 0$ for all but one of the channels – call the non-zero channel $d$ – and replace $i_d$ with its quasi-active, zero-frequency approximation (*Equation 15*). We choose a

spatially uniform holding potential $v_h$ and expansion point $(y_1^0, \ldots, y_K^0)$ for ion channel $d$. We absorb constant terms in $i_i$ and obtain (*Equation 21*) in terms of $\delta v_i = (v_i - v_h)$:

$$g_{Li}\,\delta v_i + \sum_{n \in \mathcal{N}_i} g_{C,in}\,(\delta v_i - \delta v_n) + \overline{g}_{di} l_d(v_h) \delta v_i = i_i. \tag{25}$$

We again recast this equation in matrix form, but distinguish the known passive component $G_{\mathrm{pas}}$ from the unknown diagonal matrix $G_{v_h,\mathrm{chand}}$ containing as its $i$'th diagonal element the channel term $\overline{g}_{di} l_d(v_h)$:

$$\left(G_{\mathrm{pas}} + G_{v_h,\mathrm{chand}}\right)\delta \mathbf{v} = \mathbf{i}. \tag{26}$$

From the biophysical model, we compute $Z$ by setting all ion channel conductances except channel $d$ to zero, to obtain the fit (*Equation 5*).

By consequence, we have a system of $N^2$ equations, linear in the $N$ unknown maximal conductances $\overline{\mathbf{g}}_d = (\overline{g}_{d1}, \ldots, \overline{g}_{dN})$, that can be recast in a form $A\,\overline{\mathbf{g}}_d = \mathbf{b}$, where $A$ and $\mathbf{b}$ depend on the holding potential and expansion points. On the one hand, we aim to compute these linearizations for a sufficiently large range of holding potentials and expansion points. On the other hand, the fit should be restricted to domains of the ion-channel phase space where the channel resides during normal input integration, so as to avoid over-fitting on domains of the phase space that are never reached. We choose four holding potentials around which to linearize: $v_h = -75, -55, -35$ and $-15$ mV. For channels with a single state variable $y$, we choose $y^0 = y_\infty(v_h)$ for these four holding potentials. For channels with two state variables, we computed $y_1^0 = y_{\infty 1}(v_h)$ and $y_2^0 = y_{\infty 2}(v_h)$ for the four holding potentials, and choose sixteen expansion points as all possible combinations of these two state variables. We then weigh the matrices $A$ and vectors $\mathbf{b}$ with the inverse of the open probability $f(\blacksquare)$ for their respective expansion point, concatenate the matrices $A$ and vectors $\mathbf{b}$ for each of these expansion points, and obtain the system $A_{\mathrm{ext}}\,\overline{\mathbf{g}}_d = \mathbf{b}_{\mathrm{ext}}$ of $EN^2$ equations (with $E$ the number of expansion points) and $N$ unknowns. We solve this system in the least-squares sense for $\overline{\mathbf{g}}_d$ and repeat this procedure for each voltage-dependent ion channel in the biophysical model.

4. Finally, we fit $e_{Li}$ $(i = 1, \ldots, N)$ to reproduce the resting membrane potential $\mathbf{v}_{\mathrm{eq}} = (v_{\mathrm{eq}1}, \ldots, v_{\mathrm{eq}N})$ of the biophysical model evaluated at the compartment sites. We substitute this potential in (*Equation 21*), set $\dot{v}_i = 0$ and $i_i = 0$ for $i = 1, \ldots, N$ and obtain:

$$g_{Li}\left(v_{\mathrm{eq}i} - e_{Li}\right) + \sum_{n \in \mathcal{N}_i} g_{C,in}\left(v_{\mathrm{eq}i} - v_{\mathrm{eq}n}\right) + \sum_{c \in \mathcal{C}_i} i_c(v_{\mathrm{eq}i}) = 0, \quad i = 1, \ldots, n. \tag{27}$$

This is a system of $N$ equations, linear in the $N$ unknowns $e_{Li}$ $(i = 1, \ldots, N)$, and can thus be solved with standard algebraic techniques.

## Synaptic weight-rescale factors

To be able to analytically compute the weight-rescale factor when a synapse with temporal conductance $g(t)$ is moved from its original site $s$ to a compartment site $c$ on a dendritic tree, we use the steady-state approximation (*Equation 18*). Because we implicitly convolve the rescaled synaptic current with the temporal input impedance kernel $z_{cc}(t)$ at the compartment site when we simulate the reduced model, the weight-rescale factor still yields accurate temporal voltage responses (*Figure 4B,E,H*). In the original configuration, we have

$$\begin{pmatrix} v_c \\ v_s \end{pmatrix} = \begin{pmatrix} cc z_{cc} & z_{cs} \\ z_{sc} & z_{ss} \end{pmatrix} \begin{pmatrix} i_c \\ g(t)\,(e - v_s) \end{pmatrix}, \tag{28}$$

with $i_c$ an arbitrary current at the compartment site and $g$ resp. $e$ the synaptic conductance resp. reversal. Eliminating $v_s$ from this yields

$$v_c = \frac{\left(z_{cc} + (z_{cc} z_{ss} - z_{cs} z_{sc})\,g(t)\right) i_c + z_{cs}\,g(t)\,e}{1 + z_{ss}\,g(t)}. \tag{29}$$

In the reduced configuration we have:

$$v'_c = z_{cc} \left( i_c + \beta\, g(t)\, (e - v'_c) \right). \tag{30}$$

From requiring that $v_c = v'_c$, $\beta$ is found as follows:

$$\beta = \frac{z_{cs}}{z_{cc}} \frac{z_{cs} i_c - e}{\left[ 1 + \left( z_{ss} - \frac{z_{cs}^2}{z_{cc}} \right) g(t) \right] z_{cc} i_c - [1 + (z_{ss} - z_{cs}) g(t)] e}. \tag{31}$$

When $z_{cc} \approx z_{cs}$, which is true in basal dendrites and a reasonable approximation in many apical dendrites if $c$ is on the direct path from $s$ to the soma, (*Equation 31*) reduces to

$$\beta = \frac{1}{1 + (z_{ss} - z_{cc})\, g(t)}. \tag{32}$$

The weight-rescale factor hence depends on time. Decomposing the time-varying $g(t)$ into $g + \delta g(t)$, with $g$ the temporal average and $\delta g(t)$ the fluctuations around $g$, we find (*Equation 8*) and (*Equation 9*) from requiring that the denominator of (*Equation 32*) be constant in time:

$$\beta = \frac{1}{1 + \Delta z\, g + \Delta z\, \delta g(t)} = \frac{1}{(1 + \Delta z\, g)\left( 1 + \frac{\Delta z\, \delta g(t)}{1 + \Delta z\, g} \right)} \approx \frac{1}{1 + \Delta z\, g} \quad \text{if} \quad \frac{\Delta z\, \delta g(t)}{1 + \Delta z\, g} \ll 1 \tag{33}$$

where $\Delta z = z_{ss} - z_{cc}$.

In the case of an NMDA synapse, the conductance depends sigmoidally on the local voltage. Using scale factor (*Equation 8*), we obtain for the rescaled synaptic current at the compartment site:

$$i'_s = \frac{g\, \sigma(v_s; v_T, \Delta_v)}{1 + (z_{ss} - z_{cc})\, g\, \sigma(v_s; v_T, \Delta_v)} (e - v_c). \tag{34}$$

This current still depends on the voltage at the synaptic site $v_s$ in the sigmoid. A current at the compartment site that causes a voltage $v_c$ there would have caused a voltage

$$v_s = v_{eq} + \frac{z_{ss}}{z_{cc}} (v_c - v_{eq}) \tag{35}$$

if it was placed at the synapse site. Hence, to retain the same activation level of the NMDA synapse, we substitute (*Equation 35*) in the sigmoid. This amounts to a change in threshold and width of the sigmoid:

$$\sigma(v_s; v_T, \Delta_v) \longrightarrow \sigma\left( v_{eq} + \frac{z_{ss}}{z_{cc}} (v_c - v_{eq}); v_T, \Delta_v \right) = \sigma\left( v_c; \frac{z_{cc}}{z_{ss}} (v_T + v_{eq}) - v_{eq}, \frac{z_{cc}}{z_{ss}} \Delta_v \right). \tag{36}$$

We will denote this modified sigmoid as $\sigma'(v)$. We have not yet succeeded in reformulating (*Equation 34*) by only rescaling its parameters with constant rescale factors, since $1/(1 + (z_{ss} - z_{cc})\, g\, \sigma'(v_c))$ still depends on time through its dependence on $v_c$. To obtain constant rescale factors, we substitute the (*Equation 34*) in (*Equation 17*):

$$v_c - v_{eq} = z_{cc}(t) \star \left[ \frac{g\, \sigma'(v_c)}{1 + (z_{ss} - z_{cc})\, g\, \sigma'(v_c)} (e - v_c) \right]. \tag{37}$$

Here, the current in square brackets is the synaptic current. The convolution to obtain $v_c$ is implemented implicitly by the compartmental model. We then approximately eliminate the denominator:

$$v_c - v_{eq} \approx z_{cc}(t) \star \left[ g\, \sigma'(v_c)\, (e - v_c) - \frac{z_{ss} - z_{cc}}{z_{cc}} g\, \sigma'(v_c)(v_c - v_{eq}) \right] \tag{38}$$

to obtain for the synaptic current:

$$i'_s = \frac{z_{ss}}{z_{cc}} g\, \sigma'(v_c) \left( \frac{z_{cc}}{z_{ss}} (e - v_{eq}) + v_{eq} - v_c \right). \tag{39}$$

We thus find that the synaptic weight is rescaled by a constant factor $\frac{z_{ss}}{z_{cc}}$ and the reversal potential is shifted to $\frac{z_{cc}}{z_{ss}} (e - v_{eq}) + v_{eq}$. Note however that this reduction relies on two key assumptions: that

there are no other inputs to the dendritic tree and that there are no fluctuations in $v_c$ generated by input currents not at site $s$, as this could lead to spurious activation or suppression of $\sigma'(v_c)$.

Suppose now we shift $M$ conductance synapses to $N$ compartments. Let $\mathbf{v} = (v_1, \ldots, v_N, v_{N+1}, \ldots, v_{N+M})$ be the vector with $K = N + M$ components containing the voltages at the $N$ compartment sites and at the $M$ synapse sites. Similarly $\mathbf{g} = (0, \ldots, 0, g_1, \ldots, g_M)$ is a vector of $K$ components containing $N$ zeros and the $M$ synaptic conductances, while $\mathbf{e} = (0, \ldots, 0, e_1, \ldots, e_M)$ contains the synaptic reversals. In matrix form, (*Equation 18*) for this system becomes:

$$\mathbf{v} = Z \operatorname{diag}(\mathbf{g}) (\mathbf{e} - \mathbf{v}), \tag{40}$$

and its solution:

$$\mathbf{v} = \Gamma \mathbf{e}, \tag{41}$$

with $\Gamma$ a matrix given by:

$$\Gamma = (I + Z \operatorname{diag}(\mathbf{g}))^{-1} Z \operatorname{diag}(\mathbf{g}). \tag{42}$$

In the reduced setting, we assign each synapse to one compartment site (a reasonable choice could be the closest site). We introduce an $N \times M$ compartment assignment matrix $C$ where an element $c_{nm}$ is 1 if synapse $m$ is assigned to compartment $n$ and zero otherwise. We further introduce a vector of reduced compartment voltages $\mathbf{v}' = (v_1', \ldots, v_N')$, a vector of rescaled synaptic conductances $\mathbf{g}_\beta' = (\beta_1 g_1, \ldots, \beta_M g_M)$ and a vector of synaptic reversals $\mathbf{e}' = (e_1, \ldots, e_M)$. In the reduced model, voltages are obtained from:

$$\mathbf{v}' = Z C \operatorname{diag}(\mathbf{g}_\beta') (\mathbf{e}' - C^T \mathbf{v}'). \tag{43}$$

We then require that $\mathbf{v}_N = \mathbf{v}'$, with $\mathbf{v}_N$ a vector containing the first $N$ components of $\mathbf{v}$. We denote by $\Gamma_{NM}$ the matrix containing the first $N$ rows and last $M$ columns of $\Gamma$, and note that we can write (*Equation 41*) as $\mathbf{v}_N = \Gamma_{NM} \mathbf{e}'$. Substituting this in (*Equation 43*) yields:

$$\Gamma_{NM} \mathbf{e}' = Z C \operatorname{diag}(\mathbf{g}_\beta') (\mathbf{e}' - C^T \Gamma_{NM} \mathbf{e}'). \tag{44}$$

We then fit the parameters $\beta_1, \ldots, \beta_M$ (here absorbed in $\mathbf{g}_\beta'$) in the least-square sense from:

$$\Gamma_{NM} = Z C \operatorname{diag}(\mathbf{g}_\beta') (I - C^T \Gamma_{NM}), \tag{45}$$

which again is a linear fit.

## Experimental recordings

Coronal slices (300 μm thick) containing the anterior cingulate cortex (ACC) were prepared from 10 to 12 week old C57BL/6 mice using a vibratome on a block angled at 15 degree to the horizontal in ice-cold oxygenated artificial cerebral spinal fluid (ACSF) and then maintained in the same solution at 37°C for 15–120 min. Normal ACSF contained (in mM) NaCl, 125; NaHCO$_3$, 25; KCl, 2.5; NaH$_2$PO$_4$, 1.25; MgCl$_2$, 1; glucose, 25; CaCl$_2$, 2; pH 7.4. Individual neurons were visualized with a Nikon Eclipse E600FN fit with a combination of oblique infrared illumination optics and epifluorescence, the switch between optical configurations was software-triggered (*Sieber et al., 2013*). Pyramidal neurons were selected on the clearly visible, proximal apical dendrite. This selection criterion resulted in a homogeneous population of pyramidal neurons based on their firing properties and shape of the AP (i.e. all cells possessed a prominent after-hyperpolarization and a significant sag ratio at the soma). Dual somatic and dendritic whole-cell patch-clamp recordings were performed from identified L5 pyramidal neurons in the rostroventral ACC (1.1–1.4 mm below the pial surface, 1.1–0.2 mm rostral to the Bregma) using two Dagan BVC-700. During the experiments, the external recording solution (normal ACSF) was supplemented with 0.5 mM CNQX and 0.5 mM AP-5 to block excitatory glutamatergic synaptic transmission. Experiments were performed at physiological temperatures between 34–37°C. Whole-cell recording pipettes (somatic, 4 to 8 MΩ; dendritic, 12 to 32 MΩ), were pulled from borosilicate glass. The internal pipette solution consisted of (in mM) potassium gluconate, 135; KCl, 7; Hepes, 10; Na$_2$-phosphocreatine, 10; Mg-ATP, 4; GTP, 0.3; 0.2% biocytin; pH 7.2 (with KOH); 291–293 mosmol l$^{-1}$. For somatic recordings, 10–20 μm Alexa 594 was

added to the intracellular solution: first, the soma was patched (whole-cell configuration by negative pressure); after 5 min of intracellular perfusion, the fluorescent signal allowed for the clear identification of the apical dendritic tree, then the dendritic region of interest was patched with a smaller pipette. Compensation was performed in current clamp mode by recovering the fast, initial square voltage response to a hyperpolarizing current injection (−100 pA, 50 ms). First the pipette capacitance was compensated to the level that the voltage response showed an immediate voltage drop due to the series resistance of the pipette that was adjusted subsequently. Compensation of dendritic series resistance yielded values between 30 and 60 MΩ for pipettes with a resistance between 17 and 21 MΩ. On average, series resistance was 2.3 times larger than the pipette resistance. Series resistance of both somatic and dendritic recording electrodes was monitored frequently, and experiments were terminated when proper compensation was not possible anymore (i.e. reached values of more than four times the pipette resistance). All cells were filled with biocytin, and PFA-fixed slices were developed with the avidin–biotin-peroxidase method for Neurolucida reconstructions (*Egger et al., 2008*). Data analysis was performed using Igor software (Wavemetrics) and Excel (Microsoft).

## Simulation-specific parameters

### Parameters *Figure 2*

For the sequence detection (*Figure 1f*), the synapse model is as in *Branco and Häusser, 2010*. The maximal conductance of the AMPA component is $\overline{g}_{\mathrm{AMPA}} = 0.5$ nS and its conductance window shape given by an alpha function (*Rotter and Diesmann, 1999*) with a time scale of 2 ms. The NMDA component is given by a kinetic model (*Destexhe et al., 1998*) with $\overline{g}_{\mathrm{NMDA}} = 8$ nS and external magnesium concentration of 1 mM, and receives an exponentially decaying ($\tau = .5$ ms) neurotransmitter concentration with amplitude of 5 mM. For the input-order detection (*Figure 1G*), AMPA synapses 1 and 2 use the standard parameters and have respective maximal conductances of 10 and 5 nS. For the simulation with the L5 pyramidal cell, excitatory synapses have AMPA+NMDA components with $R_{\mathrm{NMDA}} = 2$ and $\overline{g} = 3$ nS. For inhibitory synapses, $\overline{g} = 2$ nS. In the Purkinje cell, excitatory synapses only have AMPA components with $\overline{g} = 10$ nS, while for inhibitory synapses, $\overline{g} = 5$ nS. For the L2/3 pyramid, L5 pyramid, resp. Purkinje cell, excitatory Poisson rates are 2, five resp. 6 Hz and inhibitory firing rates are 4, one resp. 2 Hz. Simulation were run for 10,000 ms.

### Parameters *Figure 3*

APs are evoked with a DC current pulse. For panels A–C, pulse amplitude is $i_{\mathrm{amp}} = 0.5$ nA and pulse duration is $t_{\mathrm{dur}} = 5$ ms. For panels D–F, we have $i_{\mathrm{amp}} = 3$ nA and $t_{\mathrm{dur}} = 1$ ms. For panels G–I, $i_{\mathrm{amp}} = 1.5$ nA and $t_{\mathrm{dur}} = 5$ ms. For panels J–K, the somatic current pulse had $i_{\mathrm{amp}} = 1.9$ nA and $t_{\mathrm{dur}} = 5$ ms, while the dendritic current injection had a double exponential waveform, with $\tau_r = 0.5$ ms and $\tau_d = 5$ ms and amplitude $i_{\mathrm{amp}} = 0.5$ nA. Onset of the somatic current pulse precedes onset of the dendritic current injection by 5 ms.

### Parameters *Figure 4*

In panel B, we inject Ornstein–Uhlenbeck (OU) processes for current (mean $\mu = 0.08$ nA and standard deviation $\sigma = 0.025$ nA) and conductance ($\mu = 0.005$ μS, $\sigma = 0.0025$ μS). Reversal was 0 mV for the excitatory conductance and −80 mV for the inhibitory conductance. All OU processes had a time scale of 30 ms. In panels D–F, we use AMPA+NMDA synapses with $\overline{g} = 0.5$ nS and $R_{\mathrm{NMDA}} = 3$. Burst firing is mimicked by drawing synapse activation times from a Gaussian distribution with as mean the burst time and a standard deviation of 2 ms.

In panels G–J, AMPA+NMDA synapses at the compartment site (green square) have $\overline{g} = 1$ nS and $R_{\mathrm{NMDA}} = 2$. To determine the NMDA-spike threshold, we activate between 0 and 200 synapses in a burst. The burst is again modeled by drawing spike times from a Gaussian distribution with a 2 ms standard deviation. We then multiply the amplitude of the resulting waveform with its half-width, and average this quantity over five trials for each number of activated synapses. The number of synapses at which this quantity increases most is taken to be the NMDA-spike threshold.

We aim to test the synaptic weight-rescale factors for the AMPA and GABA synapses $\beta_{\mathrm{cond}}^{\mathrm{ext}}$ under a wide range of input conditions. To do so, we conduct simulations with different total time-averaged conductance loads $g_{\mathrm{avg}}$, exerted by the AMPA and a GABA synapse on the subtree that is to

be reduced. These synapses are activated with spike times drawn from a homogeneous Poisson process. We also also change the size of the conductance fluctuations, by conducting simulations with different firing rates for the homogeneous Poisson processes (since the total conductance load for a simulation is fixed, a small firing rate means that individual synaptic weights have to be increased to reach the desired total conductance load, hence resulting in larger conductance fluctuations). We furthermore change the number of locations on the subtree $n_{loc}$ over which the conductance load is distributed. Note that by consequence, the conductance load at each location is $g_{avg}/n_{loc}$. We also adapt these locations to a specified average difference in input resistance $\Delta z_{avg}$ between the compartment site and the location. Finally, we change the average voltage level achieved by the combined AMPA and GABA input by changing the 'nudging potential' $e_n$ (**Urbanczik and Senn, 2014**)

$$e_n = \frac{g_{AMPA} e_{AMPA} + g_{GABA} e_{GABA}}{g_{AMPA} + g_{GABA}}, \tag{46}$$

where $g_{AMPA}$ resp. $g_{GABA}$ are the time-average conductances of individual AMPA resp. GABA synapses, and $g_{AMPA} = 0$ mV resp. $g_{GABA} = -80$ mV their respective reversal potentials.

We thus draw five meta-parameters for each simulation: $g_{avg}$, $r_{avg}$, $n_{loc}$, $\Delta z_{avg}$ and $e_n$. The time-averaged conductance at each location is given by

$$g_{avg}/n_{loc} = g_{AMPA} + g_{GABA}. \tag{47}$$

Together with (**Equation 46**), this fixes the time-averaged conductance of each individual AMPA synapse and GABA synapse. Since the temporal conductance profile of a synapse (e.g. the AMPA synapse) following the arrival of a single input AP is modeled as the product of a weight factor $w_{AMPA}$ and a unitary conductance window $g_{uAMPA}(t)$, the time-averaged conductance of that synapses activated by a homogeneous Poisson process input is

$$g_{AMPA} = w_{AMPA} \, r_{avg} \int_0^{\infty} \mathrm{d}t \, g_{uAMPA}(t). \tag{48}$$

Thus, given $r_{avg}$ and $g_{AMPA}$, the synaptic weight $w_{AMPA}$ can be extracted (and similarly for the GABA synapse).

To perform simulations across a wide range of input conditions, we draw 200 samples of these five meta-parameters from the intervals: $e_n \in [-80\text{mV}, -50\text{mV}]$, $g_{avg} \in [0.01\text{nS}, 300\text{nS}]$, $r_{avg} \in [1\text{Hz}, 100\text{Hz}]$, $n_{loc} \in [1, 20]$, and $\Delta z_{avg} \in [0\text{M}\Omega, 1023.2\text{M}\Omega]$ following the Latin hypercube (LH) method (**Press et al., 2007**). $g_{avg}$ and $r_{avg}$ are sampled on a log scale, whereas for all other parameters, we use a linear scale.

## Parameters *Figure 5*

To obtain compartment sites, we divide the longest branch in each basal subtree in $n$ parts of equal length, thus giving us $n$ distances from the soma, where we increase $n$ from 0 (point-neuron) to 10. In each subtree, we distribute compartments at all sites at these distances, and also add all bifurcation sites in between compartment sites. In this way, we obtain 11 reductions that we quantify according to 'no. of compartments per 100 µm of dendrite'. We implement 400 'background' synapses; 200 AMPA and 200 GABA synapses with an average conductance of $g_{AMPA}$ resp. $g_{GABA} = 4 g_{AMPA}$ and firing rate $r_{AMPA} = r_{avg}$ resp. $r_{GABA} = r_{avg}$. We implement 20 synapse clusters, consisting of AMPA+NMDA synapses with maximal conductance $\overline{g}_{A+N}$ and $R_{NMDA} = 2$. These clusters are activated with a Poissonian burst rate $r_{burst}$, the number of spikes per burst is drawn from a Poisson distribution with parameter $n_{avg}$, and spike times are drawn for each burst time according to a Gaussian distribution with standard deviation of 5 ms. For each of the parameters not given previously, 10 LH samples were drawn from $g_{AMPA} \in [0.4\,\text{nS}, 0.8\,\text{nS}]$, $r_{avg} \in [1.5\,\text{Hz}, 3.0\,\text{Hz}]$, $\overline{g}_{A+N} \in [0.5\,\text{nS}, 1.5\,\text{nS}]$, $n_{avg} \in [10, 30]$, and $r_{burst} \in [0.25\,\text{Hz}, 0.6\,\text{Hz}]$, resulting in 10 different output spike rates shown in B. For each parameter set, a simulation was run for 10,000 ms.

## Parameters *Figure 6*

Hyper- resp. depolarizing current steps have $i_{amp} = -.3$ nA resp. $i_{amp} = .1$ nA and $t_{dur} = 500$ ms. The full model was optimized with an evolutionary algorithm using the BluePyOpt library (**Van Geit**

*et al., 2016*), where we ran 100 iterations with and offspring size of 100. Goodness of fit was evaluated in a multi-objective manner as the root mean square error of the resting voltage (average voltage 100 ms before each current step), the final step voltage amplitudes after sag (average voltage during the last 100 ms of the DC current injection), and the voltage root mean square error of the full trace. We optimized the specific membrane capacitance and the conductance densities for passive leak and $h$, $K_{ir}$ and $K_m$ channels. The membrane currents followed an exponential distribution $g(x) = g_0 e^{x/d_x}$, with $x$ the distance from the soma, and as parameters $g_0$ – the conductance at the soma – and $d_x$ – the length constant of the distribution.

## Data and software availability

NEAT (NEural Analysis Toolbox), our open-source Python toolbox to obtain reduced models, is available on https://github.com/unibe-cns/NEAT (copy archived at swh:1:rev: 1cb15f36aa0a764105348541d046c85ef38e21ee).

## Acknowledgements

This work was supported by the Swiss National Science Foundation (Grant 159872 to TN; Grant 180316 to WS – with F Helmchen), the European Union's Horizon 2020 Framework Programme (Grant 720270, 785907 and 945539 to WS – Human Brain Project SGA 1–3) and the European Research Council (Grant 682905 to TN). We thank all our lab colleagues for helpful discussions and critical comments on the figures.

## Additional information

### Funding

| Funder | Grant reference number | Author |
| --- | --- | --- |
| H2020 European Research Council | 720270 | Walter Senn |
| Swiss National Science Foundation | 180316 | Walter Senn |
| Swiss National Science Foundation | 159872 | Thomas Nevian |
| H2020 European Research Council | 785907 | Walter Senn |
| H2020 European Research Council | 945539 | Walter Senn |
| H2020 European Research Council | 682905 | Thomas Nevian |

The funders had no role in study design, data collection and interpretation, or the decision to submit the work for publication.

### Author contributions

Willem AM Wybo, Conceptualization, Resources, Software, Formal analysis, Validation, Investigation, Visualization, Methodology, Writing - original draft, Writing - review and editing; Jakob Jordan, Resources, Software, Writing - review and editing; Benjamin Ellenberger, Resources, Software; Ulisses Marti Mengual, Resources, Data curation; Thomas Nevian, Supervision, Funding acquisition, Writing - original draft, Writing - review and editing; Walter Senn, Conceptualization, Supervision, Funding acquisition, Visualization, Writing - original draft, Project administration, Writing - review and editing

### Author ORCIDs

Willem AM Wybo (iD) https://orcid.org/0000-0003-1385-4980
Jakob Jordan (iD) http://orcid.org/0000-0003-3438-5001

Thomas Nevian 🔗 http://orcid.org/0000-0001-9804-608X
Walter Senn 🔗 https://orcid.org/0000-0003-3622-0497

## Decision letter and Author response
Decision letter https://doi.org/10.7554/eLife.60936.sa1
Author response https://doi.org/10.7554/eLife.60936.sa2

## Additional files

### Supplementary files
• Transparent reporting form

### Data availability
All data generated or analysed during this study are included in the manuscript and supporting files.

The following previously published datasets were used:

| Author(s) | Year | Dataset title | Dataset URL | Database and Identifier |
|---|---|---|---|---|
| Hay E, Hill S, Schürmann F, Markram H, Segev I | 2011 | L5b PC model constrained for BAC firing and perisomatic current step firing | https://senselab.med.yale.edu/ModelDB/showmodel.cshtml?model=139653#tabs-1 | ModelDB, 139653 |
| Branco T, Clark BA, Häusser M | 2010 | Dendritic Discrimination of Temporal Input Sequences | https://senselab.med.yale.edu/ModelDB/ShowModel?model=140828#tabs-1 | ModelDB, 140828 |
| Chen XR, Heck N, Lohof AM, Rochefort C, Morel M, Rosine W, Doulazmi M, Marty S, Cannaya V, Avci HX, Mariani J, Rondi-Reig L, Vodjdani G, Sherrard RM | 2013 | Mature Purkinje Cells Require the Retinoic Acid-Related Orphan Receptor-$\alpha$ (ROR$\alpha$) to Maintain Climbing Fiber Mono-Innervation and Other Adult Characteristics | http://www.neuromorpho.org/neuron_info.jsp?neuron_name=Purkinje-slice-ageP35-2 | NeuroMorpho.org, NMO_100072 |

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
