## [Decision Letter]

**Acceptance summary:**

A general principle in systems theory is that a complicated model often reduces to a much simpler one – a fact exploited by Rall in his classic work on reduction of passive dendritic trees. However, obtaining simplified models of nonlinear dendritic integration of distributed synaptic input has remained challenging. Wybo and colleagues develop methods to achieve such a reduction systematically, offering an approach that reduces the complexity of simulations while providing insight into the factors that allow lumping of dendritic compartments into a smaller equivalent system.

**Decision letter after peer review:**

Thank you for submitting your article "Data-driven reduction of dendritic morphologies with preserved dendro-somatic responses" for consideration by *eLife*. Your article has been reviewed by two peer reviewers, and the evaluation has been overseen by a Reviewing Editor and Ronald Calabrese as the Senior Editor. The following individuals involved in review of your submission have agreed to reveal their identity: Hugh PC Robinson (Reviewer #1); Matthew F Nolan (Reviewer #2).

The reviewers have discussed the reviews with one another and the Reviewing Editor has drafted this decision to help you prepare a revised submission.

This manuscript proposes a plausible and promising tool for reducing model complexity of morphological conductance-based models, illustrated in a variety of situations including complex simulations and experimental data fitting. This methodology will likely be useful to a variety of neurophysiologists and theorists and can permit more efficient simulation while shedding light on the core parameters that matter in neural computation at the single cell level.

Reviewers agreed that the writing needs to be clearer throughout. Terms need to be defined. Implicit references to other work or to concepts need to be replaced with explicit, declarative sentences and explanations. On the technical side the paper also needs a fuller explanation of how fitting of voltage-gated conductances is performed. Finally, a fuller discussion and illustration of the limitations would be appropriate, e.g. explicitly showing cases where further reductions cannot be achieved without fundamentally changing the system. These necessarily exist in a nonlinear system so this is not a shortcoming of the method, rather an important point that some readers may not appreciate.

Full reviewer comments are included below for information.

Reviewer #1:

This study addresses an important issue in computational neuroscience: how to reduce highly-detailed biophysical models which reproduce the full morphological complexity of neurons to models which have a degree of complexity permitting them to be understood conceptually, and simulated much more efficiently, while preserving the accuracy of the original model as far as possible, and in a well-defined way. While there is a large literature on this problem for passive neurons, it is really necessary to address this problem for active/nonlinear membranes, given what we now know about voltage-dependent ion channels in dendrites. This is a problem of fitting the reduced model parameters to the output of the highly-detailed model, and the authors take a systematic and theoretically-satisfying approach to it, by matching the impedance matrices including the phenomenological impedance of voltage-gated channels which describes the reduced model to the output of the detailed model recorded at a small number of locations. This was demonstrated for neuronal models of several key active dendritic mechanisms: back-propagating sodium spikes, calcium and NMDA spikes. The authors also provide a toolbox to facilitate this process.

I have no major concerns: I found that the approach was described and illustrated very carefully and clearly, and the method's accuracy and utility were demonstrated convincingly. The motivation was well established and the writing accurate and transparent.

Reviewer #2:

The study introduces algorithms and software that aims to generate accurate reduced compartmental models of neurons. The approach appears elegant and is well validated by exemplar applications. I think it will be of interest to many neuroscientists. For the network modeler, the software provides a practical solution to the tradeoff involved in choosing neuronal models that are accurate and those that are computationally efficient. At a conceptual level the approach may be useful in addressing questions about which details of a neuron's morphology are critical for the computations that it contributes to. A caveat of my review is that I haven't had time to carefully check any of the math – it looks about right and the validation is generally good.

(1) The initial part of the Results section is important for understanding how the software works but is hard to follow. I think more care needs to be taken with defining terms clearly and explaining the logic of the approach in an accessible way.

(2) How the approach deals with voltage-gated ion channels could be made much clearer. E.g. in the subsection “A systematic simplification of complex neuron morphologies”, the text states that G_Vh,chan_ depends on the unknown maximal ion channel conductance parameters. How? Why maximal? Voltage-gated channels cannot be maximally activated at all of the voltages tested. I don't understand how to get from here to channel conductances. It could be helpful to make a figure illustrating how fits are obtained for exemplar voltage-gated conductances.

(3) The section 'Conditions under which afferent spatial connectivity motifs can be simplified' could also be written more clearly. Concerns are similar to in point 1 above.

---

## [Author Response]

[…] Reviewers agreed that the writing needs to be clearer throughout. Terms need to be defined. Implicit references to other work or to concepts need to be replaced with explicit, declarative sentences and explanations. On the technical side the paper also needs a fuller explanation of how fitting of voltage-gated conductances is performed. Finally, a fuller discussion and illustration of the limitations would be appropriate, e.g. explicitly showing cases where further reductions cannot be achieved without fundamentally changing the system. These necessarily exist in a nonlinear system so this is not a shortcoming of the method, rather an important point that some readers may not appreciate.

We have rewritten the sections that were highlighted by the reviewers, in order to explain the unclarities. We have now clearly defined the quantities that we use. We have also explained more elaborately how the voltage-gated channels are fitted. To aid with this, we have added a further panel to Figure 1—figure supplement 1. Finally, we have added a paragraph to the Discussion, where we explain possible lines of inquiry if a reduction does not reproduce dynamics of the full model that need to be retained. To illustrate these lines of inquiry, we have added a new supplementary figure (Figure 3—figure supplement 1) where we perform a similar simulation as in Figure 3J, K, but where we left out compartments in the apical trunk. This reduction is too strong and the generation of a spike-burst cannot be mimicked by the reduced model.

Full reviewer comments are included below for information.Reviewer #2:[…] (1) The initial part of the Results section is important for understanding how the software works but is hard to follow. I think more care needs to be taken with defining terms clearly and explaining the logic of the approach in an accessible way.

We have added substantial clarifications to this section.

(2) How the approach deals with voltage-gated ion channels could be made much clearer. E.g. In the subsection “A systematic simplification of complex neuron morphologies”, the text states that G_Vh,chan_ depends on the unknown maximal ion channel conductance parameters. How? Why maximal? Voltage-gated channels cannot be maximally activated at all of the voltages tested. I don't understand how to get from here to channel conductances. It could be helpful to make a figure illustrating how fits are obtained for exemplar voltage-gated conductances.

We have explicitly described the elements of the matrix G_vh_ in the manuscript, so that the parameters that are fitted are now clear. In the Materials and methods – “Quasi-active channels” subsection, we show how the linearized channel currents that go in the matrix G_vh_ are obtained. It can be seen that they are the product of a ‘maximal conductance parameter’ with a factor that follows from the linearization, and that determines the fraction of that maximal conductance that is open at a given vh. This factor thus changes with vh. Fitting our model simultaneously at a representative set of vh values allows the fit to find a best estimate for the maximal conductance parameter. An additional panel to Figure 1—figure supplement 1 now illustrates how the impedance matrix of the full model, and the inverse of the conductance matrix for the reduced model, changes under different vh. We have furthermore added a subsection to the Materials and methods, titled ”The conductance matrix”, where we explicitly describe how Equation 3 in the main text is obtained. We believe that with our added explanations, it is now clear what the matrix G_vh_ is and how the maximal conductance parameter is fitted.

(3) The section 'Conditions under which afferent spatial connectivity motifs can be simplified' could also be written more clearly. Concerns are similar to in point 1 above.

We have added additional explanations to this section, more accurately describing things that may have been unclear. We now more explicitly motivate this section, by stating that up until this section we only considered reductions where the synaptic inputs (or electrode current inputs) were located at the compartment sites. Here, we investigate what happens if this is not the case. We also changed the terminology from ‘impedances’ to ‘resistances’. We furthermore explicitly describe why β_curr_ is close to one (because the transfer impedance z_cs_ is close to the input impedance z_cc_ if the compartment site c is located more proximal than the synaptic input site s), with help of an additional panel to Figure 1—figure supplement 1. Finally, we have substantially extended the second-to-last paragraph of this section, and were able to describe the precise nature and motivation of our simulation experiments (Figure 4G-J) more clearly.